# Climate Change Impacts on the Hydrology of the Brahmaputra River Basin

Wahid Palash [1], Sagar Ratna Bajracharya [2,3,*], Arun Bhakta Shrestha [2], Shahriar Wahid [4], Md. Shahadat Hossain [5], Tarun Kanti Mogumder [5] and Liton Chandra Mazumder [5]

1   School of Public Policy and Global Affairs and Norman B Keevil Institute of Mining Engineering, The University of British Columbia, Vancouver, BC V6T 1Z4, Canada
2   International Centre for Integrated Mountain Development, Lalitpur 44700l, Nepal
3   Riverine Landscapes Research Laboratory, Geography and Planning, University of New England, Armidale, NSW 2351, Australia
4   Commonwealth Scientific and Industrial Research Organisation, Canberra, ACT 2601, Australia
5   Institute of Water Modelling, Dhaka 1230, Bangladesh
*   Correspondence: sbajrach@myune.edu.au or sagar.bajracharya2016@gmail.com

**Abstract:** Climate change (CC) is impacting the hydrology in the basins of the Himalayan region. Thus, this could have significant implications for people who rely on basin water for their lives and livelihoods. However, there are very few studies on the Himalayan river basins. This study aims to fill this gap by presenting a water balance for the Brahmaputra River Basin using the Soil and Water Assessment Tool (SWAT). Results show that snowmelt contributed about 6% of the total annual flow of the whole Brahmaputra, 21% of the upper Brahmaputra, and 5% of the middle Brahmaputra. The basin-wide average annual water yield (AWY) is projected to increase by 8%, with the maximum percentage increase in the pre-monsoon season. The annual snowmelt is projected to decrease by 17%, with a marked decrease during the monsoon but an increase in other seasons and the greatest percentage reduction in the upper Brahmaputra (22%). The contribution of snowmelt to AWY is projected to decrease while rain runoff will increase across the entire Brahmaputra and also in the upper and middle Brahmaputra. The impact assessment suggests that the upper Brahmaputra will be most affected by CC, followed by the middle Brahmaputra. The results can be used to support future water management planning in the basin taking into account the potential impact of CC.

**Keywords:** Brahmaputra; SWAT; hydrology; water balance; snowmelt; climate change

## 1. Introduction

### 1.1. Background

Mountains play a significant role as a source of water for about one-sixth of the world's population. Thus, any changes in the hydrology and water availability in mountain basins due to climate change (CC) in the coming decades are likely to have a significant impact [1,2]. About 800 million people living in the Indus, Ganges, and Brahmaputra River Basins in the Hindu Kush Himalayan (HKH) region rely entirely on basin water for their livelihoods and cultural and religious activities [3,4]. The current generation of climate models projects a significant increase in temperature and precipitation across the HKH by the mid-twenty-first century [5,6], but the implications of these changes for the hydrology of the HKH river basins remain highly uncertain. Part of the uncertainty comes from the variability in the future projections of the climate models themselves and part from the poor understanding of the region's hydrology [6–10].

The Himalayan river basins receive more than 70% of their annual precipitation during the monsoon or rainy season (June–September) [11] and the main snow and glacier melt contribution in May–October [12]. Rain runoff dominates monsoon hydrology due to the intensity of precipitation in the rainy season, and the relative contribution of snow and

glacier melt to the water yield at this time is low [8]. However, there is significantly less rainfall in the other seasons: post-monsoon (October–November), winter and dry season (December–March), and pre-monsoon (April–May) [13]. Baseflow and non-rain runoff components such as snow and glacier melt at these times dominate basin hydrology. Thus, the people in the river basins—both in the mountains and valleys upstream and the plains downstream—depend heavily on water generated from baseflow and snow and glacier melt during non-monsoon seasons [14].

In recent years, several studies have been conducted on snow and glacier melt hydrology in the high-altitude watersheds of the Himalayan river basins [6–10,12,15–19]. Some of these also looked at the potential impact of CC. These studies have helped to fill in some of the key knowledge gaps in understanding the hydrology of the Himalayan river basins and the potential response to CC. For example, Cruz et al. [5] and Immerzeel et al. [6] suggested that there could be a significant impact on hydrology in the greater Himalayas in the coming decades due to global warming and other anthropogenic pressures. Many experts believe that the shrinking snow and glacier coverage and increasing frequency of glacier outburst floods in the HKH region over the past decade are linked to global warming. Immerzeel et al. [8] found that the greater Himalayan glaciers are retreating and losing mass at rates probably equal to those in other parts of the world. Moreover, Immerzeel et al. [8] and Lutz et al. [9] projected a rise in streamflow in the Himalayan rivers and a recession of glaciers, with the net glacier runoff increasing at least until the mid-twenty-first century before decreasing.

Although the studies of snow and glacier hydrology at high altitudes are useful in indicating the potential for future change, there have been very few hydrological and water balance studies that have quantified the contribution of the baseflow and snow and glacier melt to the seasonal water yield in the Himalayan river basins, taking into account both the high-altitude snow and glacier watersheds and low-altitude rainfed watersheds. Thus, it is not yet possible to assess the potential impact of climate change (CC) on basin hydrology as a whole and baseflow and snowmelt in particular. The main reason given for the lack of whole basin studies is the absence of measured data and, thus, numerical modeling [16]. Therefore, the present study aimed first to quantify the monthly, seasonal, and annual water balance of snow and rainfed watersheds in the whole of the Brahmaputra River Basin; second, to assess the base water availability on a regional scale; and third, to assess the potential impact of CC on the major components of hydrology and water balance (i.e., precipitation, evaporation, snowmelt, baseflow, interflow, and surface runoff). The ArcSWAT 2012 version of the Soil and Water Assessment Tool (SWAT) was used to investigate the basin hydrology and water balance over a baseline period (1998–2007). For the CC study, two major global greenhouse gas emission scenarios were taken into consideration—the representative concentration pathways (RCP) with radioactive forcing at 4.5 $Wm^{-2}$ (RCP 4.5) and 8.5 $Wm^{-2}$ (RCP 8.5) [20]—each with four climatic conditions projected to emerge by 2050. We considered 1998–2007 as the base or reference period for the Brahmaputra base hydrology analysis in this study, assuming that the 1998–2007 hydrology resembles the historic average hydrology of the basin. Therefore, any change in hydrology due to the applied CC scenario will be reported as a relative change or departure from this base hydrology of the basin for 1998–2007.

The first part of the paper describes the background of the study and the physical setup of the Brahmaputra basin. This is followed by a description of the methods and tools, a presentation of the results of the Brahmaputra SWAT model, and a detailed discussion of the basin's base hydrology and possible impacts of CC. Finally, the key findings are summarized and the prospects for the results contributing to the development of a basin-wide water management policy and planning are discussed.

### 1.2. Study Area

The Brahmaputra River Basin is one of the largest river basins in the world. It extends across parts of four countries—China, India, Bhutan, and Bangladesh (Figure 1)—and

has a drainage area of 573,000 km$^2$ [21]. The river originates from the great glacier mass of Chemayungdung in the Kailash range of Southern Tibet, China, at an elevation of 5300 m above sea level (asl) and flows 1995 km from west to east through China, 983 km northeast to southwest through the Arunachal and Assam States in India, and 230 km to the south through Bangladesh before joining with the Ganges River near Aricha in Central Bangladesh. The river has many names along this stretch, especially Yarlung Zangbo in Chinese, Tsangpo in Tibetan, and Jamuna in Bengali. The combined flow of the Brahmaputra and Ganges flows 110 km southeast as the Padma meets with the Meghna River near Chandpur and flows a further 140 km south before emptying into the Bay of Bengal [22]. The land cover in the Brahmaputra Basin is 44% grassland, 14.5% forest, 14% agricultural land, 12.8% a mosaic of cropland and natural vegetation, 11% snow and ice, 2.5% barren/sparsely vegetated land, 1.8% water bodies, 0.05% permanent wetland, and 0.02% urban area [23]. Analyzing Moderate-resolution Imaging Spectroradiometer (MODIS) images from 2002 to 2012, Barman and Bhattacharjya [24] gave a recent update on the snow and ice area of the basin with the average seasonal variation, such as 12% in January, 15% in April, 3% in July, and 5% in October.

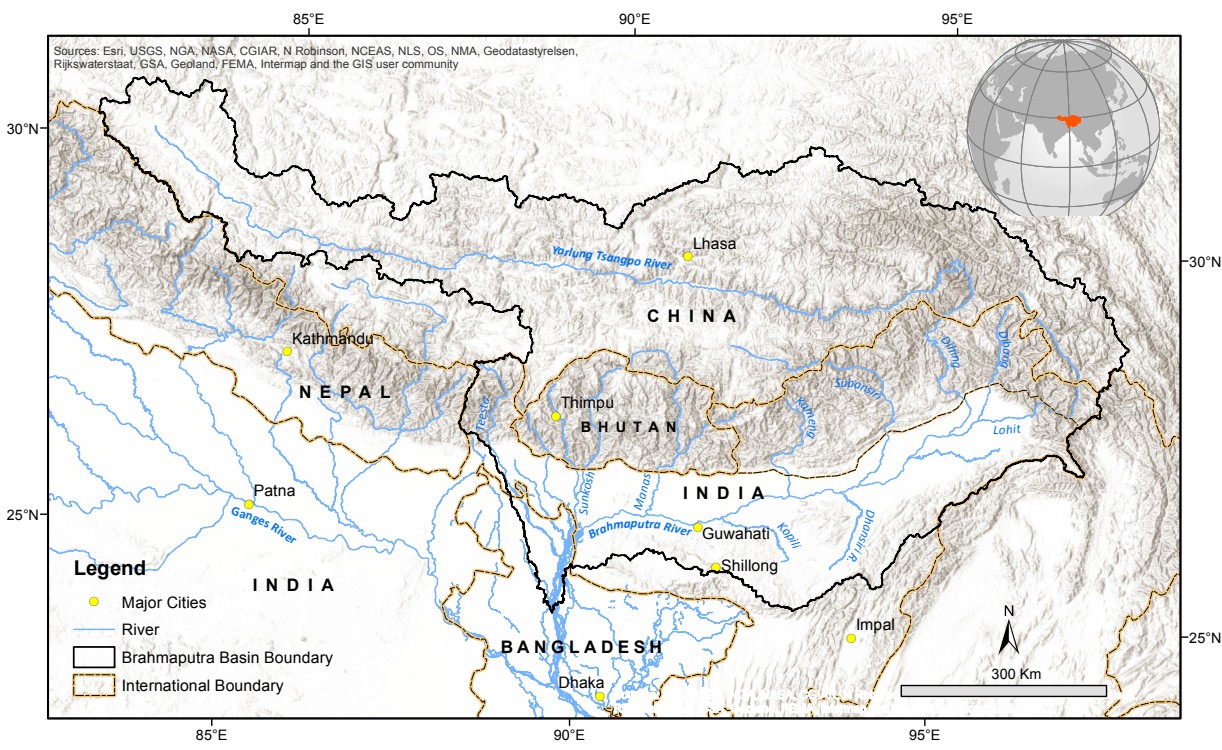

**Figure 1.** The Brahmaputra River Basin (map source: prepared by the authors).

The basin falls into three major physiographic zones: the Tibetan Plateau (>3500 m asl, 44.4% of the total basin area), the Himalayan belt (100–3500 m asl, 28.6%), and the floodplains (<100 m asl, 27%) [25]. The basin along the Southern Himalayan slope is dominated by the South Asian monsoon, with an average annual rainfall of 1400–6000 mm (average 2300 mm). In contrast, the upper part of the basin in the Tibetan Plateau (Northern Himalayan slope) has an average annual rainfall of only 300–1200 mm (average 750 mm) [23,26]. About 70–80% of the annual rainfall occurs during the monsoon season (June–September), 15–20% during the pre-monsoon period (April–May), and the remaining 5–10% in the post-monsoon (October–November) and winter and dry season (December–March) [26]. The average annual discharge of the Brahmaputra measured at Bahadurabad in Bangladesh, i.e., the gauging station furthest downstream, is about 20,200 m$^3$s$^{-1}$, and the average peak flood discharge is 70,000 m$^3$s$^{-1}$ [21].

## 2. Methods and Materials

### 2.1. Methods

The employed methods and procedures are summarized as follows:

SWAT model development: A semi-distributed hydrological model for the Brahmaputra basin capable of dealing with both surface and groundwater processes was developed using the ArcSWAT version 2012 modeling tool (http://swat.tamu.edu/ (accessed on 30 June 2020)). The model was calibrated and validated against available measured and model data for important tributaries of the basin. The developed model provides estimates of hydrology, water balance, and water availability in the basin at a sub-basin to large watershed and regional scale.

Selection of climate change (CC) scenarios: Immerzeel and Lutz [27] and Lutz et al. [28] selected CC scenarios and corresponding Global Circulation Models (GCMs) appropriate for the Hindu-Kush and Himalayan (HKH) region to investigate the climate change impact on high altitude Himalayan hydrology. We utilized the same selection in this study too. In the studies cited above, four CC scenarios were selected for each greenhouse emission scenario (RCP 4.5 and 8.5). The four CC scenarios represent four possible climate conditions—dry and cold, dry and warm, wet and cold, wet and warm—that could emerge by 2050. Details are available in Section 2.4.1.

Adjustment of climate change (CC) data: The weather data of climate change SWAT simulation was prepared by applying the projected change in temperature and precipitation under each selected CC scenario to temperature and precipitation data of the base hydrology analysis period (i.e., 1998–2007). The applied method is usually referred to as the 'delta method' where a projected temperature anomaly and percentage change in precipitation are respectively adjusted to the base temperature and precipitation dataset. Details are available in Section 2.4.2.

Scenario simulation: Nine scenarios were simulated with the Brahmaputra SWAT model: the base hydrology scenario and eight CC scenarios. We did not apply the method of continuous model simulation for a long period of time, say for 1960–1991 for base hydrology simulation or 2021–2050 for climate change hydrology simulation with continuous change in climatic variables (e.g., temperature and precipitation) or land use, land cover. Rather we simulated our model for relatively a short period of time such as 1998–2007, which we considered our base hydrology period assuming that this base hydrology resembles GCMs' reference period, that is 1960–1991, hydrology. Therefore, GCMs projected change in temperature and precipitation for the 2021–2050 period compared to the 1960–1991 period was applied to the base period data of the current study which is 1998–2007.

Analysis: We divided the Brahmaputra basin into three major regions based on simple physiographic features: the upper Brahmaputra (includes all the sub-basins on the Tibetan Plateau), the middle Brahmaputra (includes all the snow and glacier-fed sub-basins on the southern Himalayan slopes), and the lower Brahmaputra (includes all the non-snow sub-basins on the southern Himalayan slopes, the Brahmaputra floodplains, and the northern Meghalaya slopes) (Figure 2). We also calculated results separately for the large independent watersheds in the middle and lower Brahmaputra basins; e.g., Buri Dihing, Dhansiri, Dharla, Dibang, Dudhkumar, Kameng, Kopili, Lohit, Manas, Kulsi, Noa Buri Dihing, lower Subansiri, upper Subansiri, Sunkoshi, and Teesta (Figure 2). The results for the eight CC scenarios were compared with the results for the base period, and ranges of relative changes were summarized to illustrate the CC impacts at monthly, seasonal, and annual timescales.

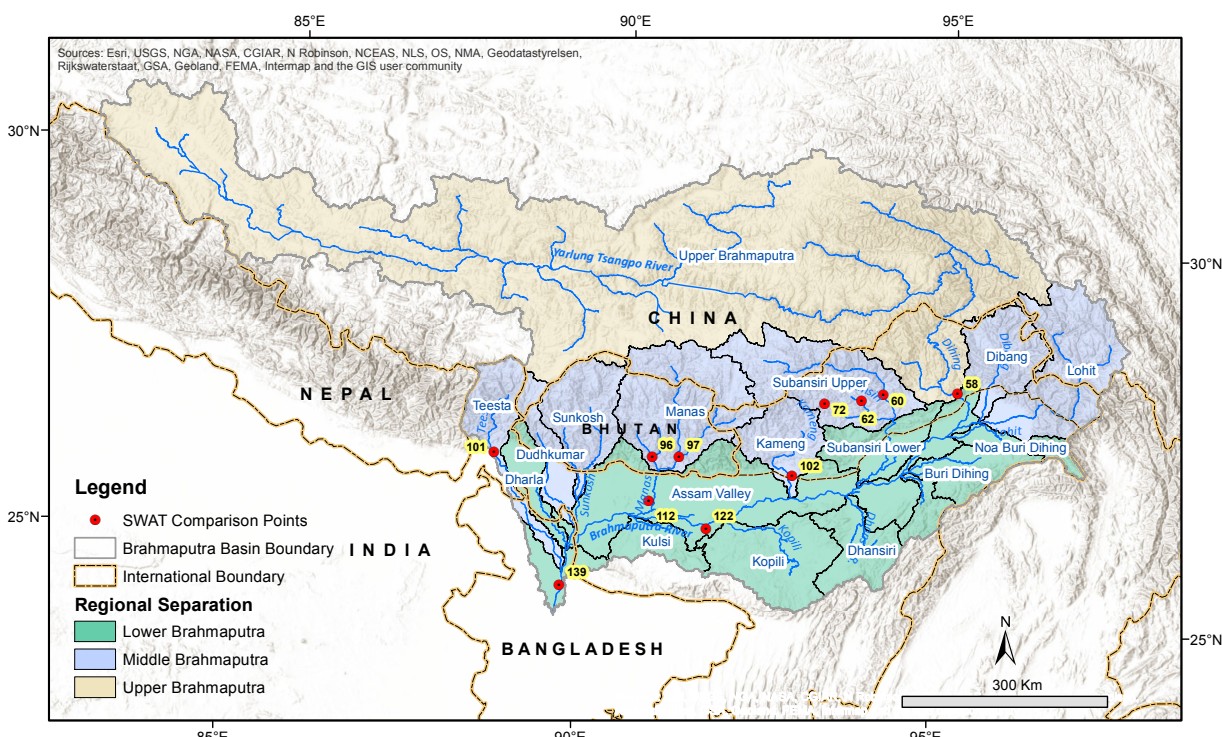

**Figure 2.** The upper, middle, and lower Brahmaputra; major watersheds in the middle and lower Brahmaputra; and positions of SWAT streamflow comparison points (map source: prepared by the authors).

### 2.2. SWAT Modeling Tool

SWAT is a physically based semi-distributed hydrologic modeling tool widely used worldwide to simulate the quality and quantity of surface and groundwater [29–35]. The model can predict the impact of land management practices or changing climatic conditions on the water, sediment, and agricultural chemical yields over long periods [36,37]. SWAT is a public domain model jointly developed by the USDA Agricultural Research Service (USDA-ARS) and Texas A&M AgriLife Research.

Setting up a SWAT model includes steps such as delineation of the watershed using land elevation data, land use and soil data processing, definition of hydrologic response unit (HRU) distribution, processing input weather data (e.g., temperature, precipitation, solar radiation, humidity, and wind speed), model simulation, calibration, and validation. The SWAT model applies the following mass balance equation to simulate the hydrology within a sub-basin:

$$SW_t = SW_0 + \sum_{i=1}^{n} \left( R_{day} - Q_{surf} - E_a - w_{seep} - Q_{gw} \right) \tag{1}$$

where $SW_t$ = final soil water content (mm), $SW_0$ = initial soil water content (mm), $t$ = time in days, $R_{day}$ = amount of precipitation on day $i$ (mm), $Q_{surf}$ = amount of surface runoff on day $i$ (mm), $E_a$ = amount of evapotranspiration on day $i$ (mm), $w_{seep}$ = amount of percolation on day $i$ (mm), and $Q_{gw}$ = amount of return flow on day $i$ (mm). The movement of water, sediments, etc. is taken place through the channel network of the watershed to the outlet.

To accommodate the snowmelt (i.e., melted water from snowfall and snow cover) dynamics into the hydrological analysis, SWAT classifies precipitation as rain or freezing rain (i.e., snow) by comparing mean daily temperature with user-defined air temperature threshold. The derived water equivalent of the snow precipitation is added to the snowpack (i.e., snow cover) and following mass balance, the equation is applied at the HRU scale [38]:

$$SNO_i = SNO_{i-1} + P_{s,i} - E_{sub,i} - SNO_{mlt,i} \tag{2}$$

where $SNO_i$ and $SNO_{i-1}$ are the water equivalents of snowpack on the current day *i* and previous day *i−1* *respectively*, $P_{s,i}$ is snow precipitation on the day *i*, $E_{sub,i}$ is the snow sublimation on the day *i*, and $SNO_{mlt,i}$ is snowmelt on day *i*. All these variables are counted as equivalent water depth (mm $H_2O$) over the total HRU area. The snowpack increases with additional snowfall but decreases with snowmelt or sublimation [39] where part of the estimated daily potential evapotranspiration is allowed to be lost by sublimation [40].

Snowmelt is controlled by the air and snowpack temperature, the melting rate, and the areal coverage of snow. Melted snow is treated the same way as rainfall for estimating runoff and percolation assuming that snowmelt occurs uniformly for a 24-h duration [38]. The snowmelt in SWAT is calculated as a linear function of the difference between average snowpack-maximum air temperature and snowmelt threshold temperature by using the following equation (Ibid):

$$SNO_{mlt} = b_{mlt} SNO_{\text{cov}} \left[ \frac{T_{snow} + T_{\max}}{2} - T_{mlt} \right] \qquad (3)$$

where $b_{mlt}$ is the melt factor (mm $H_2O$ day$^{-1}$ °C$^{-1}$), $SNO_{cov}$ is the fraction of HRU area covered by snow, $T_{snow}$ is the snowpack temperature (°C), $T_{mx}$ is the daily maximum air temperature (°C) and $T_{mlt}$ is the base temperature above which snowmelt is allowed (°C). The snowpack temperature is a function of the mean daily temperature of any given day and its previous day and is controlled by a lagging factor, $\lambda_{sno}$, such as in the following equation:

$$T_{snow,i} = T_{snow,i-1}(1 - \lambda_{sno}) + \bar{T}_{air,i} \times \lambda_{sno} \qquad (4)$$

where $T_{snow,i}$ is the snowpack temperature on a given day (°C), $T_{snow,i-1}$ is the snowpack temperature on the previous day (°C), $\bar{T}_{air,i}$ is the mean air temperature on a given day (°C) and $\lambda_{sno}$ is the snow temperature lag factor. SWAT allows up to 10 elevation bands to be defined in each sub-basin to account for the orographic effects on precipitation and temperature. The precipitation and temperature of each band are adjusted as a function of the respective lapse rate and the difference between gauge elevation and average elevation specified for the band (Ibid).

Altogether, there are seven parameters which control the snowpack accumulation and melt at a sub-basin scale [40]: the snowpack temperature lag factor TIMP ($\lambda_{sno}$) that dictates how quickly the snowpack temperature is affected by air temperature; the snowmelt base temperature SMTMP ($T_{mlt}$), above which the snowpack melts; the maximum and minimum temperature-index snowmelt factors SMFMX ($b_{mlt,mx}$) and SMFMN ($b_{mlt,mn}$); the snowfall temperature threshold SFTMP ($T_{snow}$), below which the total precipitation is taken as snow; and the areal snow coverage thresholds at 50% and 100%, SNO50COV and SNOCOVMX, that together control the areal depletion curve accounting for variable snow coverage. A detailed description of SWAT's technical background is available in Lévesque et al. [40], Neitsch et al. [38], and Wang and Melesse [39]. Details of the temperature index and elevation band approach are available in Omani et al. [41] and Rahman et al. [42].

### 2.3. Data

We applied publicly available global datasets to develop the Brahmaputra SWAT model. The Shuttle Radar Topography Mission (SRTM) generated Digital Elevation Model (DEM) of 90 m resolution [43] was used to delineate the sub-basins and river network. Land use data from GlobCover 2009 [44] and soil data from the Harmonized World Soil Database (HWSD) [45] were applied to incorporate land use and soil information for the basin (Figure 3). Weather data from the National Centers for Environmental Prediction (NCEP) Climate Forecast System Reanalysis (CFSR) [46] were used as climate input data, including precipitation, temperature, relative humidity, solar radiation, and wind speed (https://swat.tamu.edu/data/cfsr (accessed on 30 June 2020)). Satisfactory (NSE ≥ 0.5) to very good (NSE ≥ 0.65) results for simulated versus observed flow data were reported by Dile et al. [47] and Fuka et al. [48] when applying CFSR data in watershed modeling.

We also applied Moderate-resolution Imaging Spectroradiometer (MODIS) to generate monthly snow coverage [49], and the Advanced Microwave Scanning Radiometer-Earth Observing System (AMSR-E) generated monthly snow water equivalent [50] data to provide snow/glacier initial conditions in the model. Table 1 provides a summary of the data applied in the model.

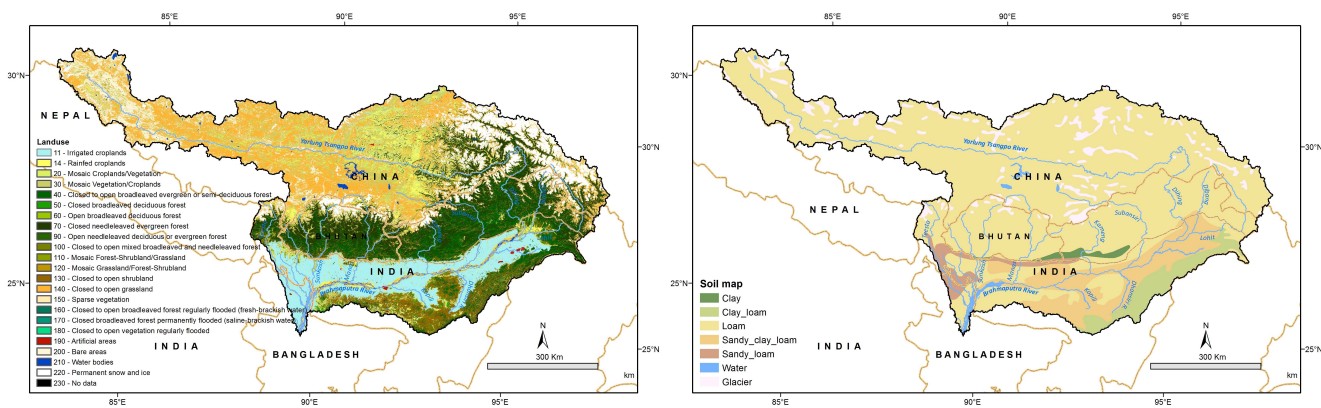

**Figure 3.** Land cover and soil map of the Brahmaputra Basin. Map source: Authors.

**Table 1.** Model input data and sources.

| Data Type | Source | Spatial Resolution | Temporal Resolution |
|---|---|---|---|
| Digital elevation model (DEM) | SRTM | 90 m | |
| Land use | GlobCover 2009 V2.3 | 1/360° | |
| Soil class map | Harmonized World Soil Database V1.2 | 1/120° | |
| Hydro-meteorological data (rainfall, temperature, relative humidity, solar radiation, wind speed) | Climate Forecast System Reanalysis (CFSR) | 0.5° | Daily |
| Weather generator data | Climate Forecast System Reanalysis (CFSR) | 0.5° | Historical analysis |
| Monthly snow cover data | MODIS/Terra Snow Cover Monthly L3 Global CMG, V5.0 | 0.5° | Monthly |
| Monthly snow water equivalent | AMSR-E/Aqua Monthly L3 Global Snow Water Equivalent EASE-Grids, V2.0 | 25 km | Monthly |
| Streamflow observations | BWDB, RivDis, ICIMOD, Immerzeel et al., [8] | | Daily |

We collected streamflow data for eleven river points along the Brahmaputra River and various tributaries representing the outflow from different sub-basins to calibrate and validate the Brahmaputra SWAT model. Details of the various points and data sources are provided in Table 2 and the spatial positions are shown in Figure 2. The Bahadurabad point—located 76.5 km downstream from the India–Bangladesh border (point 139)—is the most downstream gauging location in the basin with continuous water level and flow measurement and provides values for the outflow from the whole basin. Monthly flow measurements for this point were obtained from the Bangladesh Water Development Board (BWDB). Discharge data for points 102, 112, and 122 were obtained from the Global River Discharge (RivDIS) website (https://daac.ornl.gov/RIVDIS/rivdis.shtml (accessed on 30 June 2020)). Simulated discharge data for points 58, 60, 62, 72, 96, 97, and 101 from the Himalaya Spatial Processes in Hydrology (HI-SPHY) model [51] were obtained from the International Centre for Integrated Mountain Development (ICIMOD).

**Table 2.** Model comparison points and data availability for observed streamflow.

| River Point | Location Name | Longitude (deg.) | Latitude (deg.) | Upstream Sub-Basin Name | Data Type | Available Data Period | Data Source |
|---|---|---|---|---|---|---|---|
| 58 | Dihang | 91.88 | 29.28 | All Tibetan sub-basins in the upper Brahmaputra | | 1956—1982 | RivDis |
| 60 | | 94.15 | 28.08 | Upper Subansiri | Simulated results from HI-SPHY model | | |
| 62 | | 93.84 | 27.98 | Middle Subansiri 1 | | | |
| 72 | | 93.30 | 27.92 | Middle Subansiri 2 | | 1998—2007 | Lutz and Immerzeel [51] |
| 96 | | 90.84 | 27.01 | Manas West | | | |
| 97 | | 91.23 | 27.03 | Manas East | | | |
| 101 | Anderson Bridge | 88.54 | 26.82 | Upper Teesta | | | |
| 102 | Jai Bhorelli | 92.89 | 26.93 | Kameng | | 1958—1979 | |
| 112 | Manas | 90.84 | 26.41 | Manas (upper and lower) | Measured | 1955—1974 | RivDis |
| 122 | Pandu | 91.70 | 26.13 | Upper Brahmaputra, Dibang, Lohit, Subansiri, Kameng, Buri Dihing, Dhansiri, upper Assam Valley | | 1956—1979 | |
| 139 | Bahadurabad | 89.66 | 25.18 | Whole Brahmaputra | | 1979—2014 | BWDB |

*2.4. Climate Change Scenarios*

2.4.1. Selection of CC Scenarios and GCMs

For climate change impact assessment, we maintained the same selection of RCP scenarios and GCMs that were carried out in earlier studies such as Immerzeel and Lutz [27] and Lutz et al. [28]. In those studies, two RCP scenarios (i.e., RCP 4.5 and 8.5) with four GCMs of each representing four climatic conditions (i.e., total eight GCMs) were selected. RCP 4.5 considers a less extreme future climate with radioactive forcing stabilizing at an emission rate of 4.5 Wm$^{-2}$ by 2100. RCP 8.5, on the other hand, is a more extreme condition with radioactive forcing stabilizing at an emission rate of 8.5 Wm$^{-2}$ [20]. These scenarios were generated by Climate Model Inter-comparison Project (CMIP5) (http://cmip-pcmdi.llnl.gov/cmip5/data_portal.html (accessed on 30 June 2020)) and reported in its fifth assessment report and further discussed in Déqué et al. [52], Moss et al. [53] and Stocker et al. [54].

Immerzeel and Lutz [27] and Lutz et al. [28] tested a total 43 GCMs for RCP4.5 and 41 for the RCP8.5 scenario. For each model run, the normal annual difference in temperature and precipitation for future 2021–2050 climate conditions over a reference period 1961–1990 was determined in terms of temperature anomaly ($\Delta K$ or $\Delta T$) and percentage change of precipitation ($\Delta P$), respectively. Based on the 10th and 90th percentile values of these projected changes, four combinations of climatic conditions—dry and cold, dry and warm, wet and cold, and wet and warm—were derived for each RCP and models that gave closest results to these percentile values were selected. Selected models were then downscaled to spatial resolution 0.25° × 0.25° by using the first-order bilinear spline interpolation technique for the HKH region and monthly average projected change (i.e., delta or $\Delta$) of temperature and precipitation for each model were calculated [55].

2.4.2. Generation of Climatic Variables for CC SWAT Simulation

Under the present study, raster (gridded) data files of monthly CC projected change in temperature and precipitation for the HKH region were collected from ICIMOD, Nepal and delta for each temperature and precipitation data point utilized in the Brahmaputra SWAT model development was estimated. The rainfall and temperature data for CC SWAT simulation were then prepared by applying this delta (projected change in temperature and precipitation) to the base precipitation and temperature data of 1998–2007. The above-mentioned method, usually referred to as the delta method, is widely used in regional and local CC studies [9,55–57] and is considered an efficient way to assess climate change with multiple GCM outputs [55]. Table 3 shows the selected CC models, corresponding climatic conditions, and average projected change in temperature and precipitation over

the Brahmaputra basin. A spatial distribution of average projected changes under RCP 4.5 and 8.5 across the Brahmaputra is shown in Figure 4.

**Table 3.** Selected CC models, climatic conditions, and average projected precipitation and temperature change over the whole Brahmaputra basin.

| Model Number | Selected Model | Climate Type | RCP | ΔP (% change) | ΔT (°C anomaly) | Source |
|---|---|---|---|---|---|---|
| 1 | GISS-E2-R-r4i1p1_rcp45 | DRY, COLD | | 4.8 | 1.4 | Goddard Institute for Space Studies, NOAA, USA |
| 2 | IPSL-CM5A-LR-r4i1p1_rcp45 | DRY, WARM | RCP 4.5 | 7.2 | 2.0 | Institut Pierre Simon Laplace, France |
| 3 | IPSL-CM5A-LR-r3i1p1_rcp45 | WET, COLD | | 6.7 | 2.4 | Institut Pierre Simon Laplace, France |
| 4 | CanESM2-r4i1p1_rcp45 | WET, WARM | | 11.4 | 2.3 | Canadian Centre for Climate Modelling and Analysis, Canada |
| 5 | GFDL-ESM2G-r1i1p1_rcp85 | DRY, COLD | | 7.3 | 1.7 | Geophysical Fluid Dynamics Laboratory, NOAA, USA |
| 6 | IPSL-CM5A-LR-r4i1p1_rcp85 | DRY, WARM | RCP 8.5 | −5.5 | 2.8 | Institut Pierre Simon Laplace, France |
| 7 | CSIRO-Mk3-6-0-r3i1p1_rcp85 | WET, COLD | | 12.6 | 1.7 | Commonwealth Scientific and Industrial Research Organisation, Australia |
| 8 | CanESM2-r4i1p1_rcp85 | WET, WARM | | 12.1 | 2.8 | Canadian Centre for Climate Modelling and Analysis, Canada |

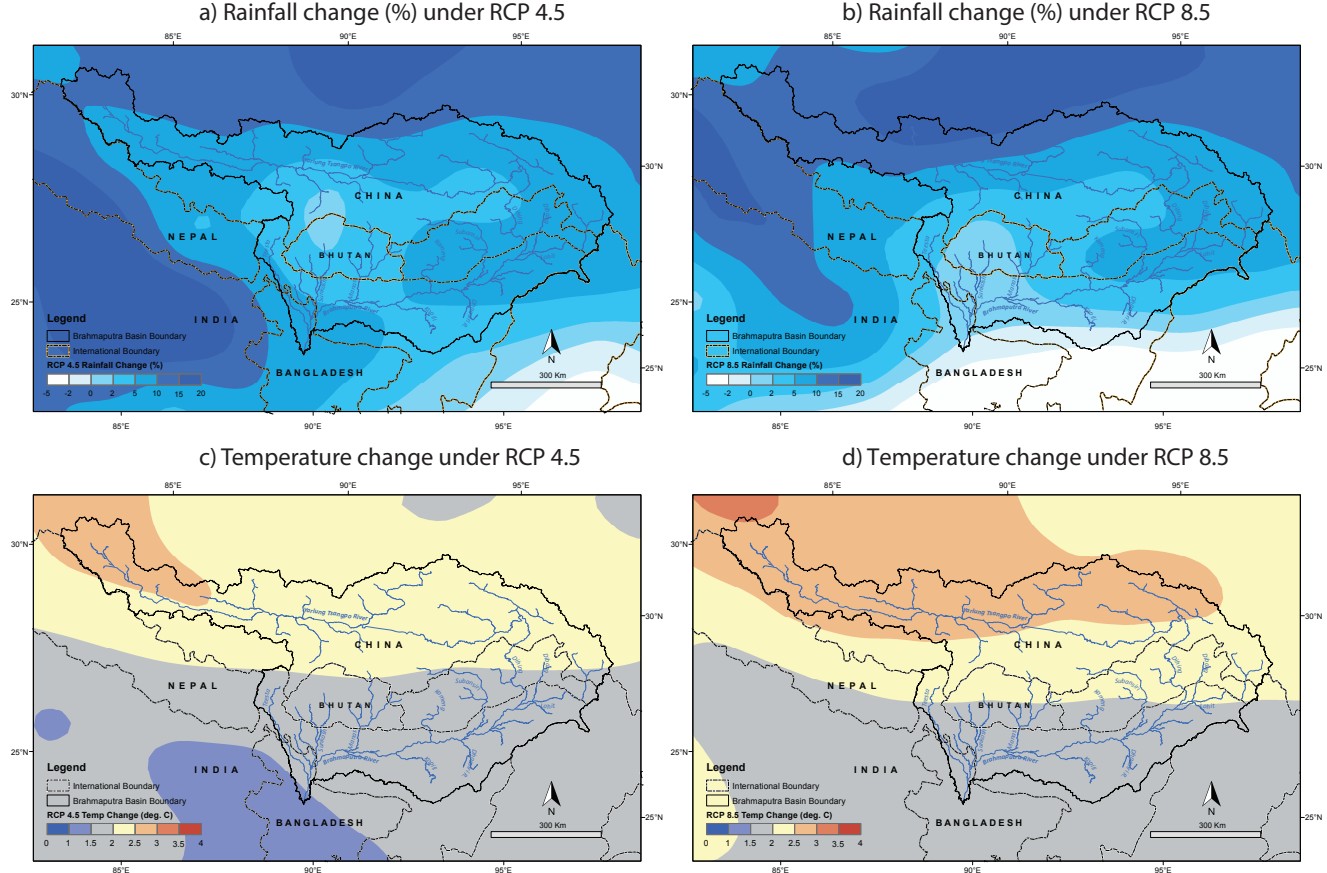

**Figure 4.** Projected precipitation and temperature change under RCP 4.5 and 8.5 climate change scenarios (mean of the four selected climatic conditions for each scenario) over the Brahmaputra River Basin.

### 2.5. Statistical Methods for Model Verification

We applied standard model verification statistical methods to verify our SWAT model results such as mean error (ME), mean absolute error (MAE), root mean square error (RMSE), RMSE–observations standard deviation ratio (RSR), coefficient of determination ($R^2$), Nash-Sutcliffe Coefficient (NSE), and percent of bias (PBIAS). A detailed description of these methods is available in Gupta et al. [58], Moriasi et al. [59], and Singh et al. [60].

## 3. Brahmaputra SWAT Model

### 3.1. Model Development

The hydrological analysis of SWAT creates 139 sub-basins in the Brahmaputra Basin after adjusting the sub-basin's threshold area and carefully customizing the stream nodes. The mean area of the sub-basins is 3747 km$^2$ with a standard deviation of 3418 km$^2$, and the largest and smallest sub-basin area is 27,826 km$^2$ and 7 km$^2$, respectively. As mentioned earlier, GlobCover land use and HWSD soil classification data are employed in the present SWAT model which creates 416,408 HRUs with a mean area of 1.25 km$^2$, maximum of 2423 km$^2$, minimum of 0.021 km$^2$ and standard deviation of 20 km$^2$. Each HRU, therefore, represents a unique land use and soil classification type, based on which SWAT defines land use and soil parameter values for the model run. Applying CFSR's daily hydro-meteorological and weather data in the SWAT model provides the key input forcing to the model such as precipitation, temperature, solar radiation, humidity, and wind speed. All the required CFSR weather data are easily downloadable from the Global Weather Data for SWAT website (https://globalweather.tamu.edu/ (accessed on 30 June 2020) in a SWAT-compatible data format which is perhaps the biggest advantage of using CFSR data in the model. We compared CFSR rainfall data with APHRODITE (Asian Precipitation—Highly-Resolved Observational Data Integration Towards Evaluation) V1101 (http://www.chikyu.ac.jp/precip/ (accessed on 30 June 2020)) and TRMM (The Tropical Rainfall Measuring Mission) 3B42 V7 (https://pmm.nasa.gov/trmm (accessed on 30 June 2020)) data in monthly and Brahmaputra regional scale (Figure 5). After a month-wise bias correction, CFSR rainfall corresponds closely to APHRODITE data giving sufficient confidence in using CFSR in our Brahmaputra SWAT mode for monthly hydrology analysis. The average spatial resolution of the current semi-distributed Brahmaputra SWAT model is 1.25 km$^2$ at the HRU scale, and 3747 km$^2$ at the sub-basin scale while the temporal resolution is one day.

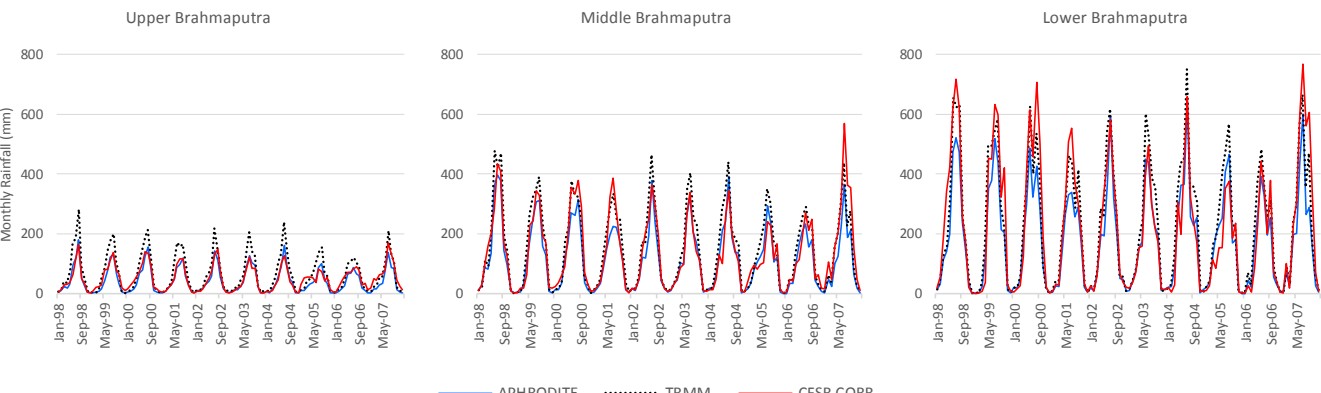

**Figure 5.** Comparison of CFSR rainfall with APHRODITE and TRMM data on monthly and regional scales.

Since the current paper aimed to simulate snowmelt hydrology in the Brahmaputra River Basin, we took special care to assign an initial condition related to snow coverage in the model setup. MODIS monthly snow cover maps were used to identify the extent of glacier coverage in different elevation bands of sub-basins while snow equivalent depths were estimated from AMSR-E data. Our model simulation starts in January 1998. Therefore,

setting snow initial condition is supposed to be a very straight-forward approach by putting MODIS derived snow coverage and AMSR-E derived snow-water depth of January 1998 in the model. However, due to uncertainty in those dataset's accuracy and to provide a typical snow condition during the month of January, we chose to apply average snow coverage and depth every January from 1998 to 2007 in our SWAT model.

*3.2. Model Calibration and Validation*

3.2.1. Simulation, Calibration and Validation Period

The Brahmaputra SWAT model was run for 1998–2013; the first seven years (1998–2004) were taken as the calibration period, and the next three years (2005–2007) as the validation period apart from Bahadurabad (i.e., river point 139) (Figure 2). The Brahmaputra discharge at Bahadurabad was validated over nine years (2005–2013). The model simulations and analysis of results were done on a monthly time-scale.

3.2.2. Challenge in Model Calibration

For a non-homogenous river basin such as Brahmaputra, it requires calibrating the model if not for every sub-basin but for a cluster of sub-basins or large watersheds to simulate a reliable result. The key difficulty of calibrating a hydrological model developed for an ungauged river basin or river basin with limited data availability such as the one we deal with here is the absence of observed streamflow data. Amid this challenge, few measured and model-estimated flow data from the earlier study were collected (see Section 2.3) and compared with our model results. Although the available data period for river points 102, 112, and 122 (Figure 2 and Table 2) do not match with the simulation period of the current SWAT model, we still compared the average monthly flow data of these points with our model results to see if our model can simulate typical streamflow for these rivers and the upstream contributing watersheds. Besides Bahadurabad and the river points mentioned above, we calibrated and validated our model at river points 58, 60, 62, 72, 96, 97, and 101 (Figure 2) by using the HI-SPHY model [51] generated streamflow data. Although using the output of another model to calibrate a new model is not necessarily a correct approach because that could easily transfer the uncertainty of earlier model results into the new model and make the new model results more uncertain. Despite this possibility, we used HI-SPHY model results to calibrate our model because of their reasonable accuracy reported in Lutz and Immerzeel [51]. Moreover, we considered that not calibrating these river points at all would induce more uncertainty in our model than calibrating with earlier model results.

3.2.3. Model Parameter Sensitivity Analysis and Calibrated Parameters

The SWAT model deals with a large number of parameters with uncertainty. Therefore, it is not often easy to calibrate the model manually. To ease out the process, the Sequential Uncertainty Fitting version 2 (SUFI2) method of the SWAT-CUP tool (http://swat.tamu.edu/software/swat-cup/ (accessed on 30 June 2020)) was applied to perform sensitivity and uncertainty analysis of model parameters and then calibrating the model. The model was initially run with the primary screening of parameters considered to be important for the watershed's rainfall-runoff and snowmelt dynamics. A series of simulations thus provide a range of sensitivity of each parameter in their simulation outputs. The sensitivity of the parameter is understood with the score of $p$-value and t-stat value. Parameters with higher absolute t-stat values and lower $p$-values are the most sensitive. The highly sensitive parameters are then considered for the second round of simulation by narrowing down the range of parameter values. The process of screening out the parameters and narrowing down the parameter range continues until a satisfied result is achieved at the calibration points. Figure 6 shows an example of the most sensitive parameters found in the Brahmaputra SWAT model and Table 4 shows a list of calibrated parameters and their adjusted values.

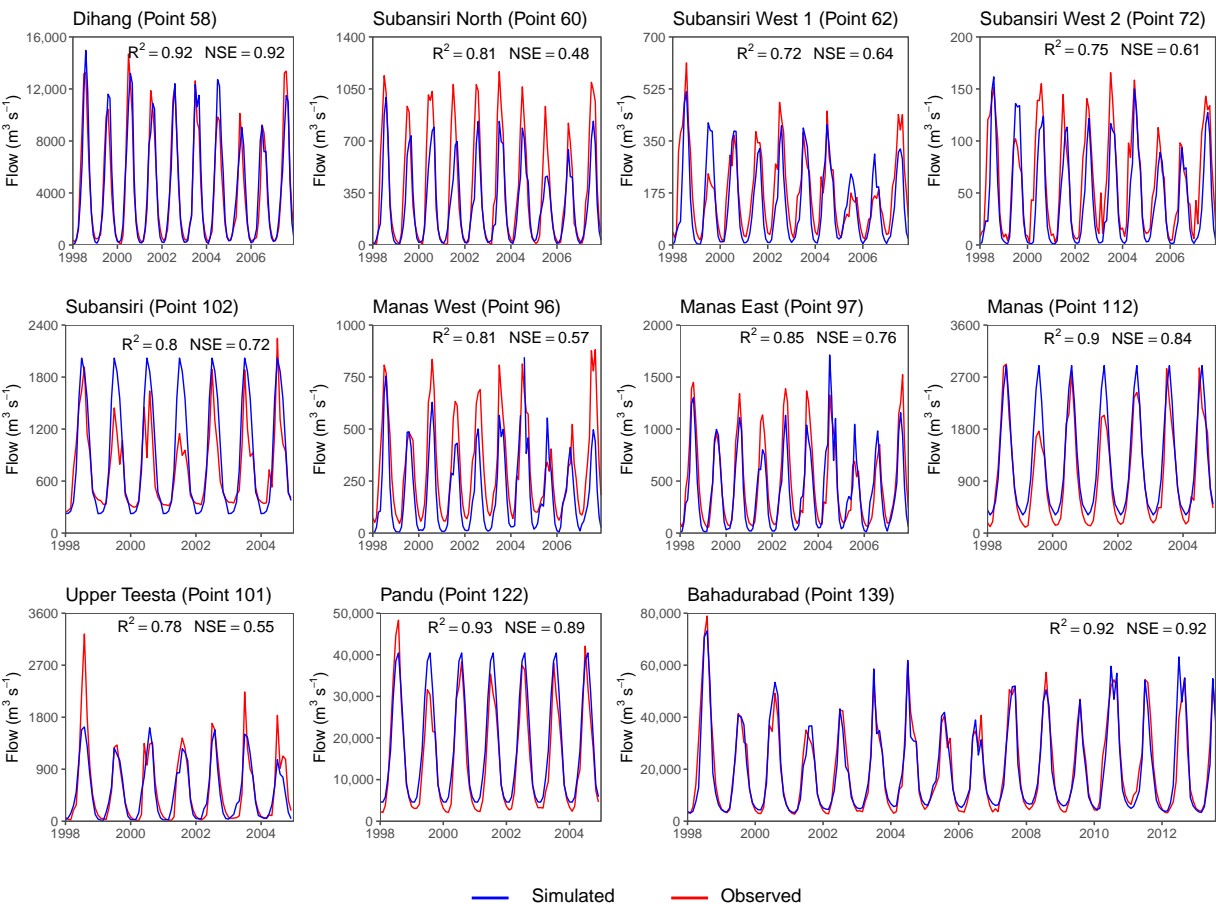

**Figure 6.** Brahmaputra SWAT model performance at the river points is shown in Figure 2. The blue line represents the observed streamflow, and the red represents the model simulation.

### 3.3. Model Performance

The reliability and efficacy of hydrological models used to study basin hydrology and water balance can be tested by comparing the similarity in observed and simulated monthly discharge rates. The observed monthly data for streamflow were compared with the monthly values simulated by the SWAT model for the period 1998–2007 at a series of river points between point 58 (Dihang), which is the outlet point of the upper Brahmaputra, and point 139 (Bahadurabad), which is the outlet point for the whole basin (results at Bahadurabad compared to the period 1998–2013). Figure 6 shows the comparison between observed and simulated discharge hydrographs at the different river points (tributaries) and Table 5 the model performance statistics at these points.

The observed and model results showed excellent agreement at most of the comparison points. For example, the PBIAS for the Dihang River was 2.9%, which indicates a slight underestimation in overall flow simulation, and the RSR was 0.32; i.e., the RMSE of the simulation is 32% of the standard deviation of the observed data. An RSR of $\leq 0.5$ is usually considered acceptable for hydrological modeling [60]. Despite the underestimation in peak discharge, the model captures the rising and falling limbs of the streamflow hydrograph very well and is particularly successful in simulating post-monsoon (October–November) and winter and dry (December–March) baseflow, and pre-monsoon (April–May) and early monsoon (June) Snowmelt. Both the value of the $R^2$ of 0.9 and the value of NSE, also 0.9, indicate a high degree of certainty in model accuracy. An NSE value $\geq 0.7$ is usually considered to indicate the very good performance of a hydrological model [58,59].

**Table 4.** Calibrated SWAT parameters with their adjusted values.

| Parameter | Value Range | Upper Brahmaputra | Middle Brahmaputra | Lower Brahmaputra | |
|---|---|---|---|---|---|
| CN2 | 35–98 | 70–87 | 53–87 | 73–84 | |
| GW_DELAY (Days) | 0–500 | 31 | 31–385 | 31–385 | |
| ALPHA_BF | 0–1 | 0.05 | 0.05–0.81 | 0.05–0.81 | |
| GWQMN (mm) | 0–500 | 1000 | 1000 | 1000 | |
| PLAPS (mm/km) | −1000–1000 | −200 | −200 | - |  |
| TLAPS (°C/km) | −10–10 | −6.5 | −6.5 | - | |
| SOL_BD (g/cc) | 0.9–2.5 | 1–1.5 | 1–1.5 | - | |
| SFTMP (°C) | −5–5 | 2 | 2 | - | |
| SMTMP (°C) | −5–5 | 2 | 2 | - | |
| SMFMX (mm $H_2O$/°C-day) | 0–4.5 | 1.5 | 1.5 | - | |
| SMFMN (mm $H_2O$/°C-day) | 0–4.5 | 0.5 | 0.5 | - | |
| TIMP | 0–1 | 0.5 | 0.5 | - | |
| SNOCOVMX (mm) | 0–500 | 200 | 200 | - | |
| SNO50COV (%) | 0–1 | 0.65 | 0.65 | - | |

CN2 = SCS runoff curve number; GW_DELAY = Groundwater delay (days); ALPHA_BF = Baseflow alpha factor (days); GWQMN = Threshold depth of water in the shallow aquifer required for return flow to occur (mm); PLAPS = Precipitation lapse rate; TLAPS = Temperature lapse rate; SOL_BD = Moist bulk density; SFTMP = Snowfall temperature; SMTMP = Snow melt base temperature; SMFMX = Maximum melt rate for snow during the year (occurs on summer solstice); SMFMN = Minimum melt rate for snow during the year (occurs on winter solstice); TIMP = Snow pack temperature lag factor; SNOCOVMX = Minimum snow water content that corresponds to 100% snow cover; SNO50COV = Snow water equivalent that corresponds to 50% snow cover.

**Table 5.** Brahmaputra SWAT model performance for different tributaries.

| River Name (Point No.) | Performance Statistics | | | | | | | | | |
|---|---|---|---|---|---|---|---|---|---|---|
| | Peak Obs. Value ($m^3s^{-1}$) | Peak Sim. Value ($m^3s^{-1}$) | Peak Error ($m^3s^{-1}$) | Mean Error (ME) ($m^3s^{-1}$) | Mean Abs. Error (MAE) ($m^3s^{-1}$) | RMSE ($m^3s^{-1}$) | $R^2$ | NSE | PBIAS (%) | RSR |
| Dihang (58) | 14,969 | 14,680 | 289 | 21 | 142 | 1326 | 0.90 | 0.90 | 2.9 | 0.32 |
| Subansiri North (60) | 994 | 1166 | −172 | −16 | 20 | 192 | 0.83 | 0.48 | 35.1 | 0.72 |
| Subansiri West 1 (62) | 516 | 613 | −97 | −6 | 10 | 82 | 0.70 | 0.62 | 24.1 | 0.62 |
| Subansiri West 2 (72) | 162 | 166 | −4 | −3 | 3 | 28 | 0.76 | 0.60 | 36.1 | 0.63 |
| Manas West (96) | 1205 | 882 | 323 | −18 | 21 | 158 | 0.75 | 0.46 | 51.7 | 0.73 |
| Manas East (97) | 1712 | 1523 | 189 | −11 | 24 | 188 | 0.82 | 0.75 | 16.1 | 0.49 |
| Upper Teesta (101) | 1631 | 3239 | −1608 | −8 | 30 | 315 | 0.78 | 0.55 | 9.9 | 0.67 |
| Kameng (102) | 2019 | 2247 | −228 | 33 | 46 | 438 | 0.67 | 0.55 | 23.2 | 0.67 |
| Manas (112) | 2899 | 2924 | −25 | 44 | 53 | 471 | 0.81 | 0.72 | 22.3 | 0.53 |
| Brahmaputra at Pandu (122) | 40,439 | 48,320 | −7881 | 619 | 718 | 6191 | 0.86 | 0.77 | 21.0 | 0.47 |
| Brahmaputra at Bahadurabad (139) | 73,266 | 78,910 | −5644 | −108 | 822 | 4889 | 0.92 | 0.92 | 2.1 | 0.29 |

The model performance in simulating the total outflow at Bahadurabad (point 139) was even better than that for the outflow from the Dihang river. Despite a slight underestimation in winter and dry season flow in most years and an overestimation of the monsoon peak in a few years, the monthly flow comparison over 1998–2013 showed very accurate model results with both $R^2$ and NSE values of 0.92. The RMSE was 4889 $m^3s^{-1}$, which is low when compared to the very high average monsoon flow (30,000 $m^3s^{-1}$). The PBIAS of 2.1% indicates a slight underestimation, while the RSR value of 0.29, which means that the RMSE is 29% of the standard deviation of the observed data, indicates a very good model estimation overall.

The model results at the other nine river points showed moderate to high accuracy. The $R^2$ and NSE value at Pandu were 0.86 and 0.77, respectively, suggesting a very good model simulation, although the PBIAS of 21% and RSR of 0.47 do not suggest a high level

of accuracy. The mean absolute error (MAE) of 718 m³s⁻¹ was relatively small compared to the monthly average flow ranging from 4577 to 40,439 m³s⁻¹. The model data showed good agreement with the estimates for total flow from the Manas basin (point 112); the RMSE, R², and NSE values of 471 m³s⁻¹, 0.81, and 0.72 respectively, indicate reasonably accurate model results, although the PBIAS of 22.3% indicates underestimation, predominantly from underestimates in the winter and dry season. The model showed only moderate performance for the tributaries in the Subansiri and Kameng Basin (points 60, 62, 72, and 102), with NSE values of only 0.46–0.62, despite high R² values of 0.67–0.83, and relatively small RMSE (28–438 m³s⁻¹) and MAE (3–46 m³s⁻¹) values. Model performance at the Teesta basin (point 101) was also moderate with RMSE, R², and NSE values of 315 m³s⁻¹, 0.78, and 0.55, respectively. The RSR value was above 0.5 although the PBIAS value indicates that the model only underestimates by 9.9%.

We compared the model results with the average annual flow of the important tributaries in the basin as estimated by the Brahmaputra Board (1995) and reported by IUCN [61] (Figures 7 and 8). There was a 1–26% anomaly between our results and IUCN-reported values. The period of the reported tributary flow (before 1995) differed from our model period (1998–2007) but we present the comparison to show the representativeness of our model in simulating annual average hydrology in the basin. Overall, the flow comparisons from different sources clearly indicate the Brahmaputra SWAT model's ability to simulate monthly, seasonal, and annual hydrology in the basin with reasonable to high accuracy.

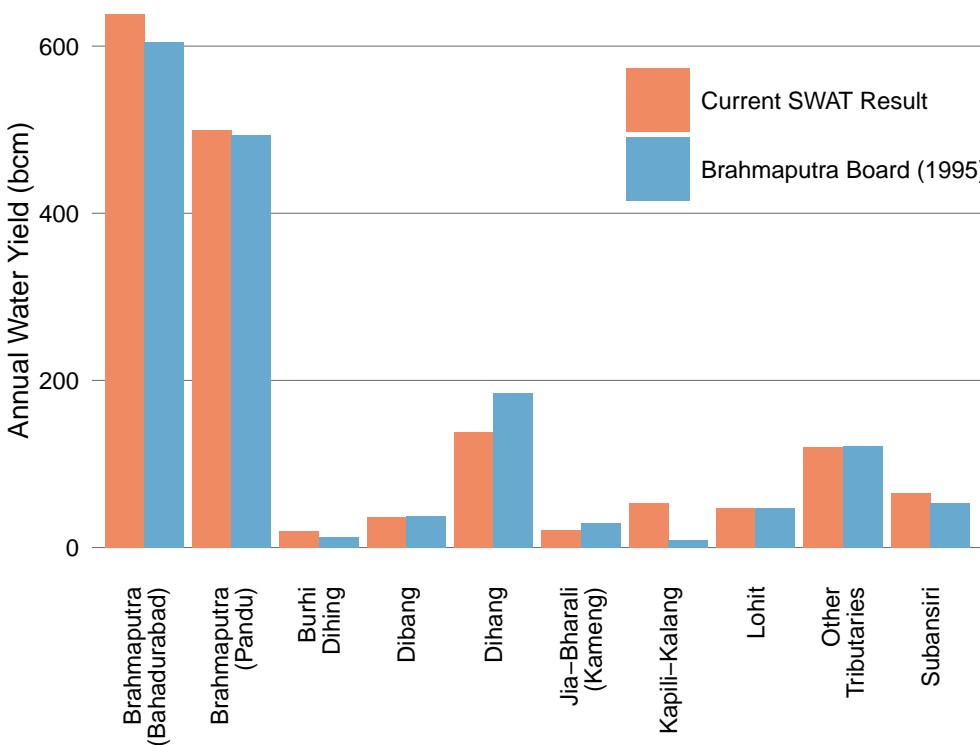

**Figure 7.** Average annual water yield (AWY) in the major tributaries of the Brahmaputra River Basin is estimated by the SWAT model and reported by the Brahmaputra Board (1995).

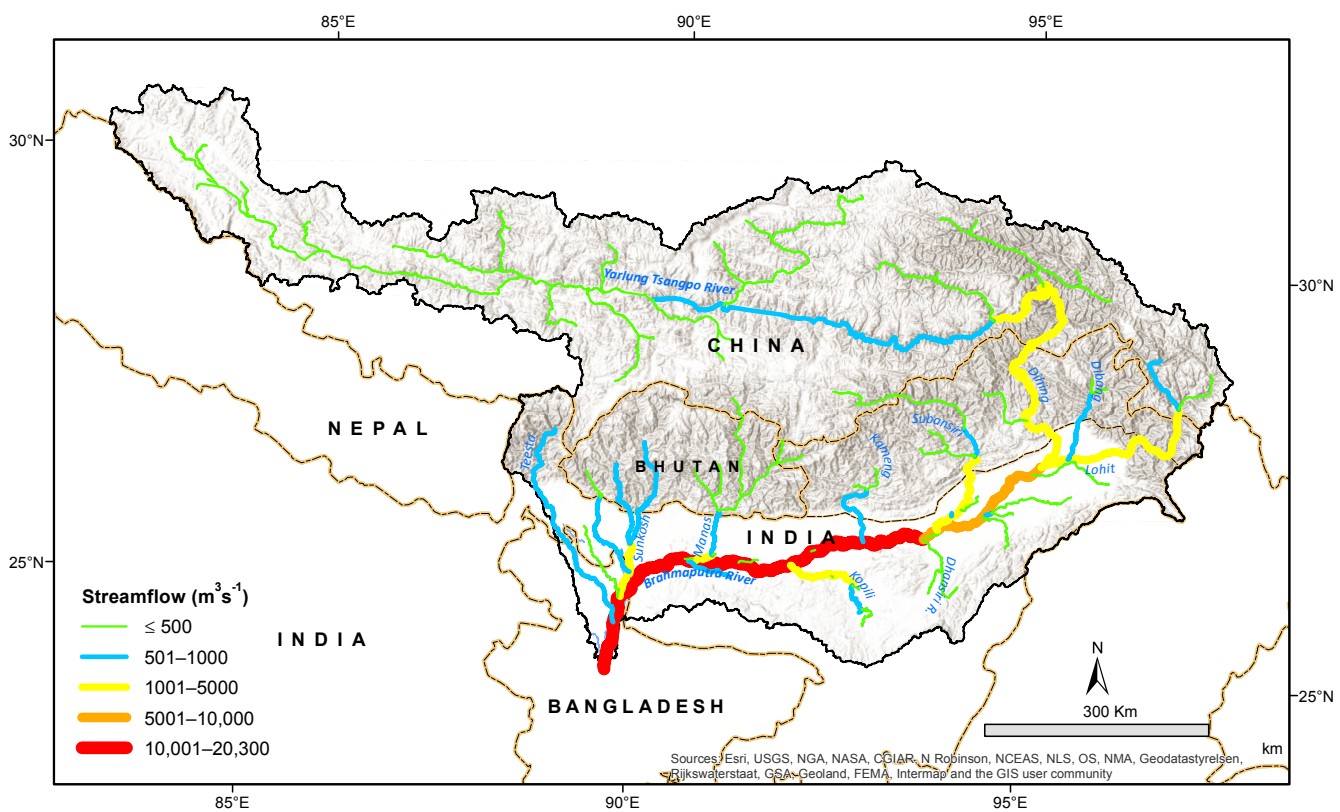

**Figure 8.** Average annual streamflow contribution by different tributaries in the Brahmaputra basin in 1998–2007.

## 4. Results and Discussion

*4.1. Hydrology and Water Balance*

4.1.1. Precipitation and Evapotranspiration

The average annual precipitation in the Brahmaputra basin in the base period (1998–2007) was estimated to be 2039 mm (Table 6 and Figure 9), with 2847 mm in the region to the south of the Himalayas (middle and lower Brahmaputra), and only 684 mm in the northern part (upper Brahmaputra)—one-third of the basin average. The lower Subansiri, Kulsi, Kopili, Buri Dihing, Sunkosh, Teesta, Dihang, and Dudhkumar watersheds had high precipitation (>3000 mm) and the Noa Buri Dihing and Dikhu watersheds had low precipitation (<1700 mm).

**Table 6.** Annual average outflow, precipitation, evapotranspiration, snowmelt, baseflow, and water yield in the base period (1998–2007).

| Name | Outflow ($m^3s^{-1}$) | Total Precipitation (mm) | Total AET (mm) | Soil Water (mm) | Snowmelt ($m^3s^{-1}$) | Surface Runoff ($m^3s^{-1}$) | Baseflow ($m^3s^{-1}$) | Interflow ($m^3s^{-1}$) | Water Yield ($m^3s^{-1}$) | Snowmelt to Runoff ($m^3s^{-1}$) |
|---|---|---|---|---|---|---|---|---|---|---|
| Brahmaputra | 21,262 | 2039 | 377 | 142 | 1911 | 10,805 | 8903 | 3122 | 22,829 | 1253 |
| Upper Brahmaputra | 4226 | 684 | 114 | 153 | 1402 | 1595 | 1775 | 1082 | 4452 | 870 |
| Middle Brahmaputra | 8122 | 2859 | 364 | 137 | 510 | 3714 | 3873 | 1949 | 9536 | 383 |
| Lower Brahmaputra | 8913 | 2837 | 684 | 120 | - | 5495 | 3255 | 381 | 8841 | - |
| Assam Valley | 19,893 | 2605 | 682 | 135 | - | 1775 | 1101 | 95 | 2971 | - |
| Buri Dihing | 676 | 3519 | 593 | 68 | - | 524 | 249 | 24 | 797 | - |
| Dhansiri | 382 | 1975 | 712 | 94 | - | 215 | 174 | 17 | 406 | - |
| Dharla | 301 | 2153 | 645 | 123 | - | 205 | 91 | 7 | 303 | - |
| Dibang | 1228 | 3270 | 332 | 121 | 121 | 423 | 555 | 266 | 1245 | 76 |
| Dikhu | 125 | 1676 | 594 | 57 | - | 53 | 57 | 16 | 126 | - |

**Table 6.** *Cont.*

| Name | Outflow (m³s⁻¹) | Total Precipitation (mm) | Total AET (mm) | Soil Water (mm) | Snowmelt (m³s⁻¹) | Surface Runoff (m³s⁻¹) | Baseflow (m³s⁻¹) | Interflow (m³s⁻¹) | Water Yield (m³s⁻¹) | Snowmelt to Runoff (m³s⁻¹) |
|---|---|---|---|---|---|---|---|---|---|---|
| Dudhkumar | 603 | 3052 | 490 | 93 | - | 300 | 209 | 96 | 604 | - |
| Kameng | 724 | 2697 | 464 | 169 | 16 | 309 | 329 | 124 | 761 | 14 |
| Kopili | 1802 | 3854 | 754 | 150 | - | 1022 | 771 | 53 | 1847 | - |
| Kulsi | 716 | 4377 | 760 | 164 | - | 408 | 287 | 21 | 717 | - |
| Lohit | 1485 | 1921 | 176 | 188 | 208 | 520 | 393 | 283 | 1195 | 173 |
| Manas | 875 | 2309 | 236 | 115 | 46 | 274 | 887 | 505 | 1667 | 31 |
| Noa Buri Dihing | 65 | 1117 | 620 | 120 | - | 36 | 24 | 7 | 66 | - |
| Lower Subansiri | 2729 | 4634 | 671 | 154 | - | 712 | 507 | 111 | 1331 | - |
| Upper Subansiri | 1178 | 2826 | 318 | 145 | 105 | 743 | 720 | 403 | 1866 | 75 |
| Sunkosh | 1179 | 3479 | 422 | 158 | 5 | 692 | 419 | 158 | 1268 | 4 |
| Teesta | 850 | 3318 | 472 | 105 | 8 | 453 | 361 | 115 | 929 | 9 |

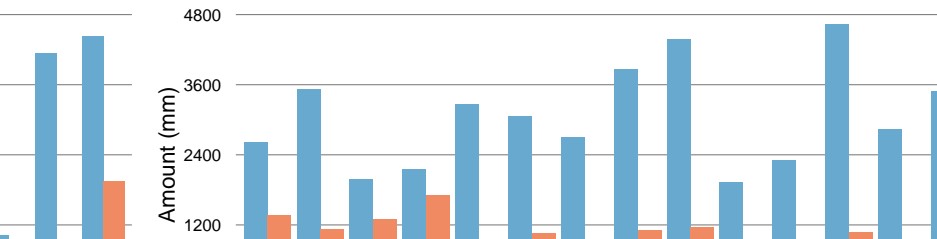

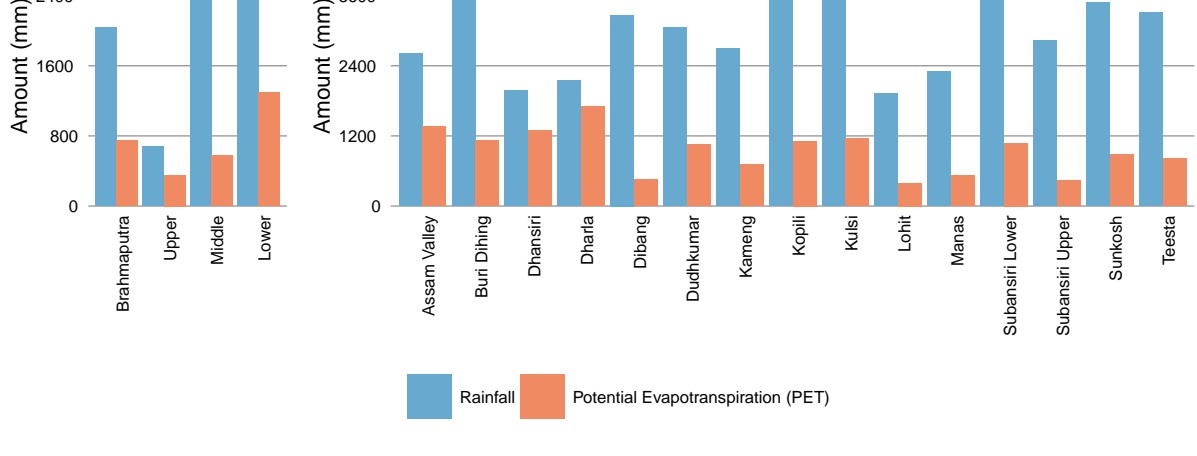

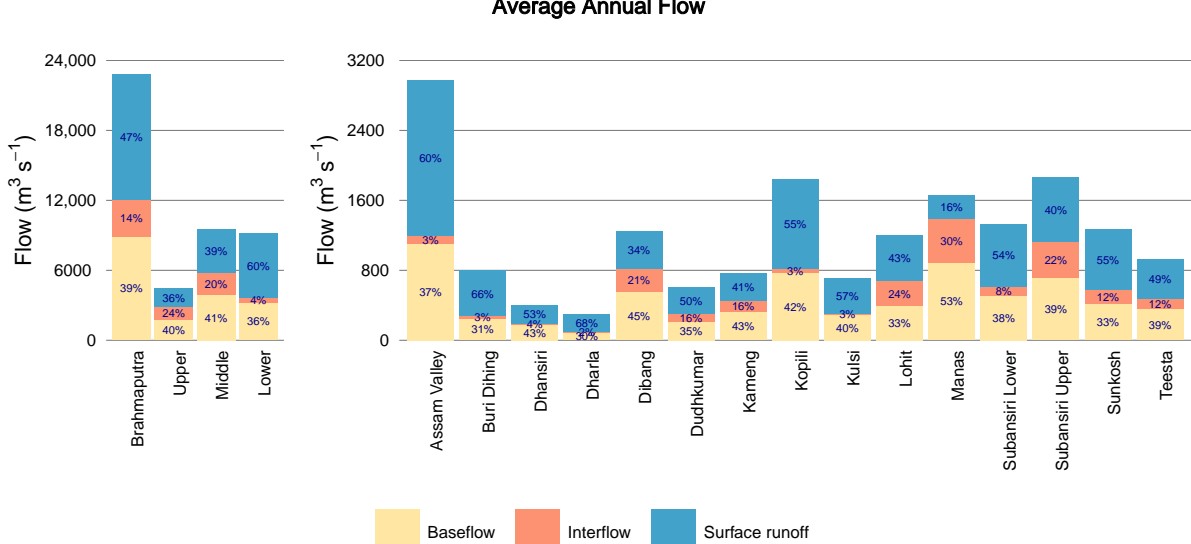

**Figure 9.** Annual average rainfall and actual evapotranspiration (top) and contribution of different flow components to annual water yield (bottom) of the Brahmaputra River Basin in 1998–2007.

The average annual potential evapotranspiration (PET) for the whole Brahmaputra basin was 750 mm (Table 6 and Figure 9) with PET to precipitation ratio was 37%. The PET was more than 1.5 and 3.5 times higher in the middle and lower Brahmaputra than in the upper Brahmaputra, respectively. The PET also showed a high spatial variation across the basin with 52% in the upper Brahmaputra, 21% in the middle and 44% in the lower Brahmaputra region. Among watersheds, the lowest PET was found in the Lohit and Manas watersheds (<300 mm) and the highest in the Kulsi, Kopili, Dhansiri, Assam Valley, lower Subansiri, and Dharla watersheds (>600 mm).

### 4.1.2. Baseflow, Water Yield and Snowmelt

The average annual water yield (AWY) and the contribution of various components over the whole Brahmaputra basin and the individual sub-basins are shown in Table 6; the monthly values for the whole Brahmaputra and upper, middle and lower the Brahmaputra in Figures 10 and 11; and seasonal values in Table 7. The AWY for the whole basin over the base period (1998–2007) was estimated to be 22,829 m$^3$s$^{-1}$, with 7% in the winter and dry season (December–March), 9% pre-monsoon (April–May), 68% monsoon (June–September), and 16% post-monsoon (October–November). Surface runoff contributes the most (47%) followed by baseflow (39%). The snowmelt contribution to basin outflow, distributed over all three components of runoff, is about 6%. The upper Brahmaputra contributes 20% of basin AWY, the middle Brahmaputra 42%, and the lower Brahmaputra 39%. Of the sub-basins, the Assam Valley contributes 13%; the Dibang, Kopili, Lohit, Manas, upper and lower Subansiri, and Sunkosh between 5.2 and 8.2% each; and the remainder less than 3.5% each.

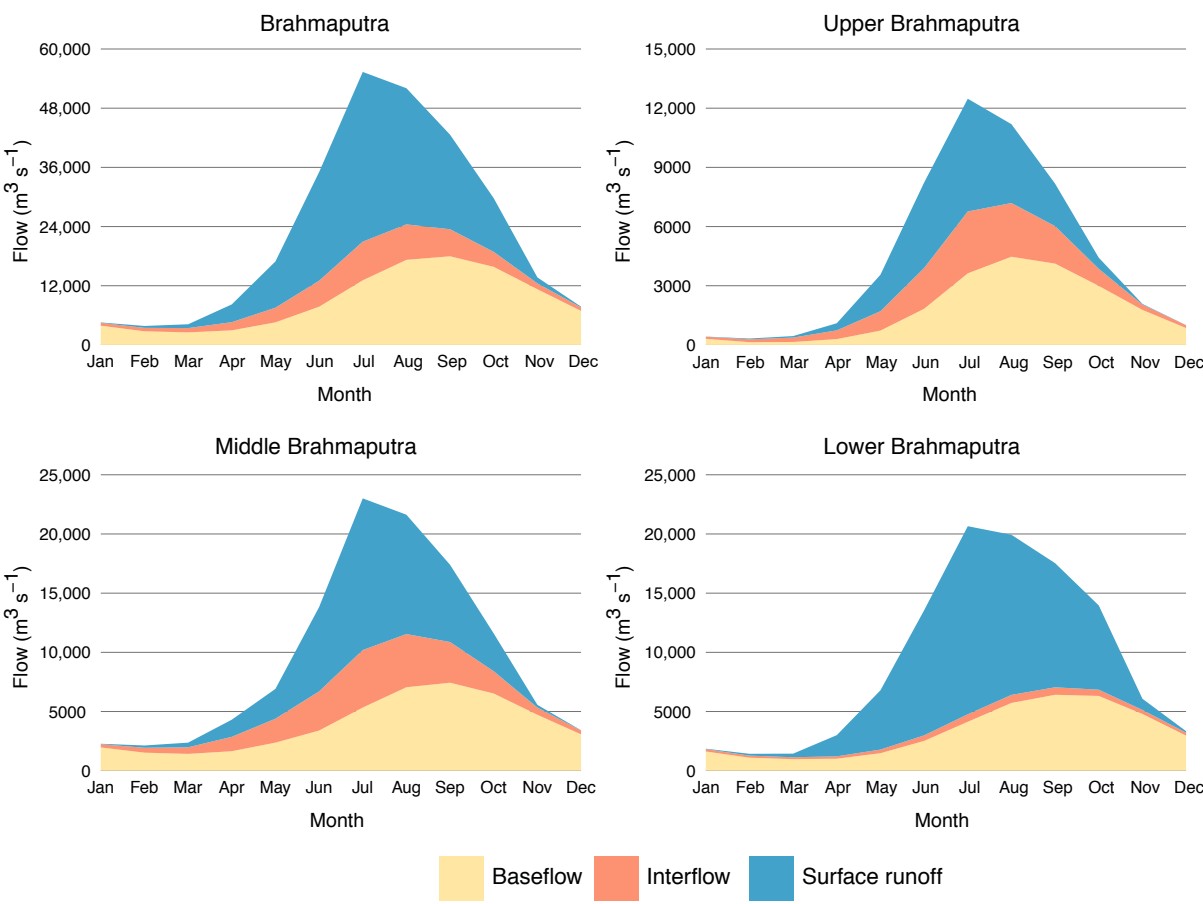

**Figure 10.** Average monthly contribution of different flow components to monthly water yield in the Brahmaputra basin and its three regions in 1998–2007.

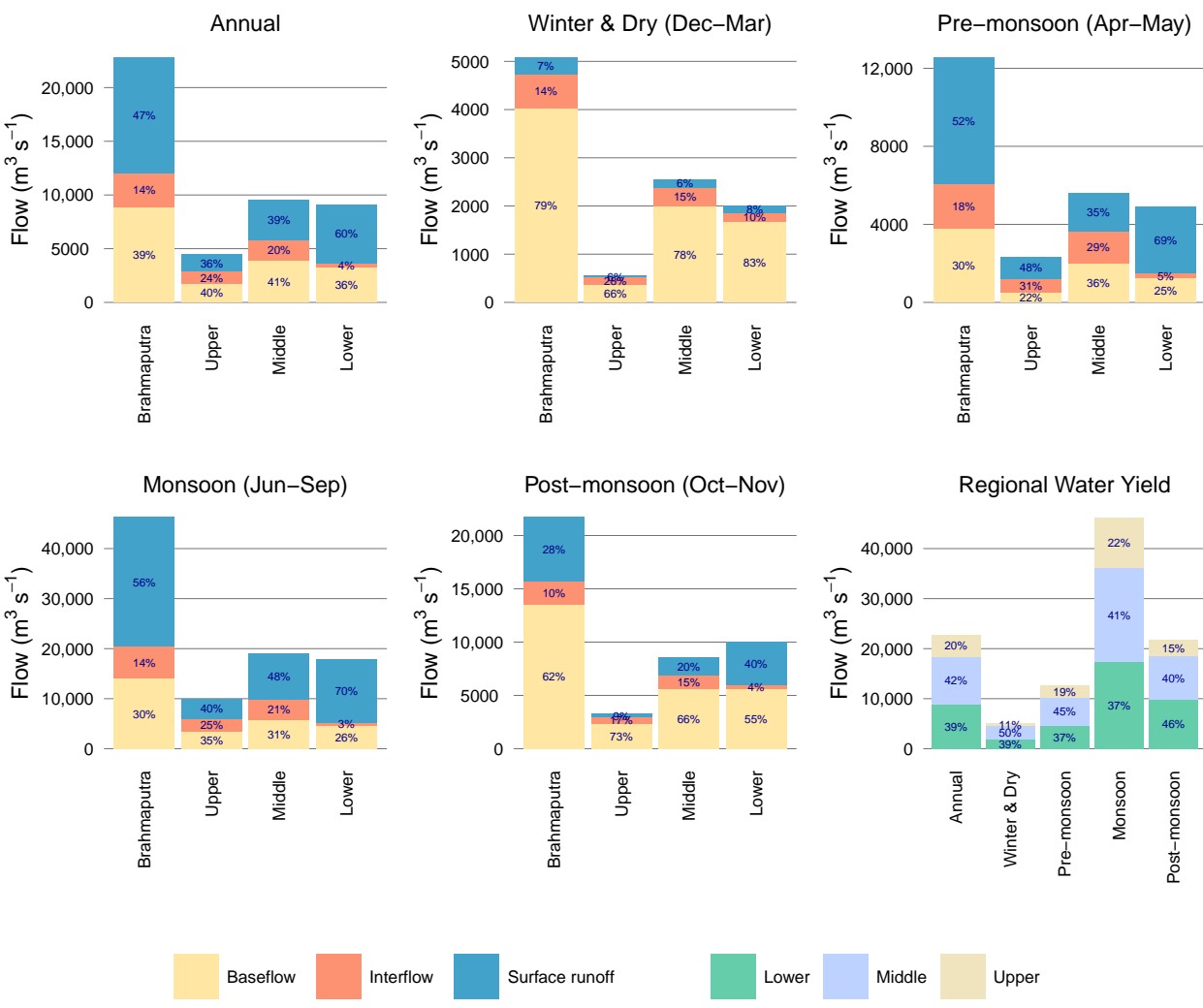

**Figure 11.** Average seasonal contribution of different flow components to seasonal water yield of the Brahmaputra basin and its three regions in 1998–2007.

**Table 7.** Seasonal average outflow, precipitation, evapotranspiration, snowmelt, baseflow, and water yield in the reference/base period (1998–2007) for the Brahmaputra as a whole and by region.

| Name | Season | Outflow (m³s⁻¹) | Total Precipitation (mm) | Total AET (mm) | Snowmelt (m³s⁻¹) | Surface Runoff (m³s⁻¹) | Baseflow (m³s⁻¹) | Interflow (m³s⁻¹) | Water Yield (m³s⁻¹) | Snowmelt to Runoff (m³s⁻¹) |
|---|---|---|---|---|---|---|---|---|---|---|
| Brahmaputra | Winter and Dry | 4384 | 388 | 144 | 150 | 348 | 4027 | 712 | 5087 | 94 |
| | Pre-monsoon | 11,906 | 1979 | 579 | 3012 | 6489 | 3770 | 2311 | 12,569 | 1311 |
| | Monsoon | 43,662 | 4189 | 569 | 4048 | 25,802 | 14,016 | 6443 | 46,261 | 2965 |
| | Post-monsoon | 19,573 | 1100 | 259 | 60 | 6039 | 13,562 | 2110 | 21,712 | 87 |
| Upper Brahmaputra | Winter and Dry | 565 | 354 | 31 | 32 | 33 | 364 | 153 | 550 | 3 |
| | Pre-monsoon | 2131 | 722 | 123 | 1902 | 1106 | 511 | 711 | 2329 | 660 |
| | Monsoon | 9459 | 1145 | 209 | 3215 | 4049 | 3514 | 2455 | 10,018 | 2269 |
| | Post-monsoon | 3178 | 384 | 82 | 15 | 298 | 2385 | 567 | 3250 | 15 |
| Middle Brahmaputra | Winter and Dry | 1870 | 529 | 157 | 118 | 161 | 1997 | 392 | 2549 | 90 |
| | Pre-monsoon | 4943 | 2386 | 592 | 1110 | 1984 | 2010 | 1617 | 5612 | 651 |
| | Monsoon | 16,291 | 6137 | 511 | 833 | 9136 | 5804 | 4023 | 18,963 | 696 |
| | Post-monsoon | 7466 | 1436 | 256 | 45 | 1709 | 5623 | 1246 | 8578 | 72 |
| Lower Brahmaputra | Winter and Dry | 1948 | 250 | 240 | - | 154 | 1666 | 197 | 1987 | - |
| | Pre-monsoon | 4833 | 2759 | 1041 | - | 3398 | 1248 | 253 | 4629 | - |
| | Monsoon | 17,912 | 6077 | 1056 | - | 12,617 | 4698 | 602 | 17,280 | - |
| | Post-monsoon | 8929 | 1614 | 473 | - | 4032 | 5555 | 434 | 9884 | - |

The proportion of baseflow and surface runoff in water yield varies significantly in different parts of the basin and in different months and seasons (Figures 10–12). For example, while the contribution of baseflow to AWY is almost identical in the upper, middle, and lower Brahmaputra at around 36–41%, the contribution of surface runoff varies from 36% in the upper Brahmaputra to 60% in the lower Brahmaputra, and the contribution of interflow from 24% in the upper Brahmaputra to only 4% in the lower Brahmaputra (Figure 11). The contribution of baseflow to AWY varies across the sub-basins from 30% in Dharla to 53% in Manas (Figure 9). The seasonal contributions of different flow components to water yield are also highly variable across the basin (Figures 10–12). About 4% of the annual flow in the upper Brahmaputra is generated in the winter and dry season, 9% pre-monsoon, 75% monsoon, and 12% post-monsoon. The corresponding contributions are 9, 10, 66, and 15% in the middle Brahmaputra, and 7, 9, 65, and 19%, in the lower Brahmaputra.

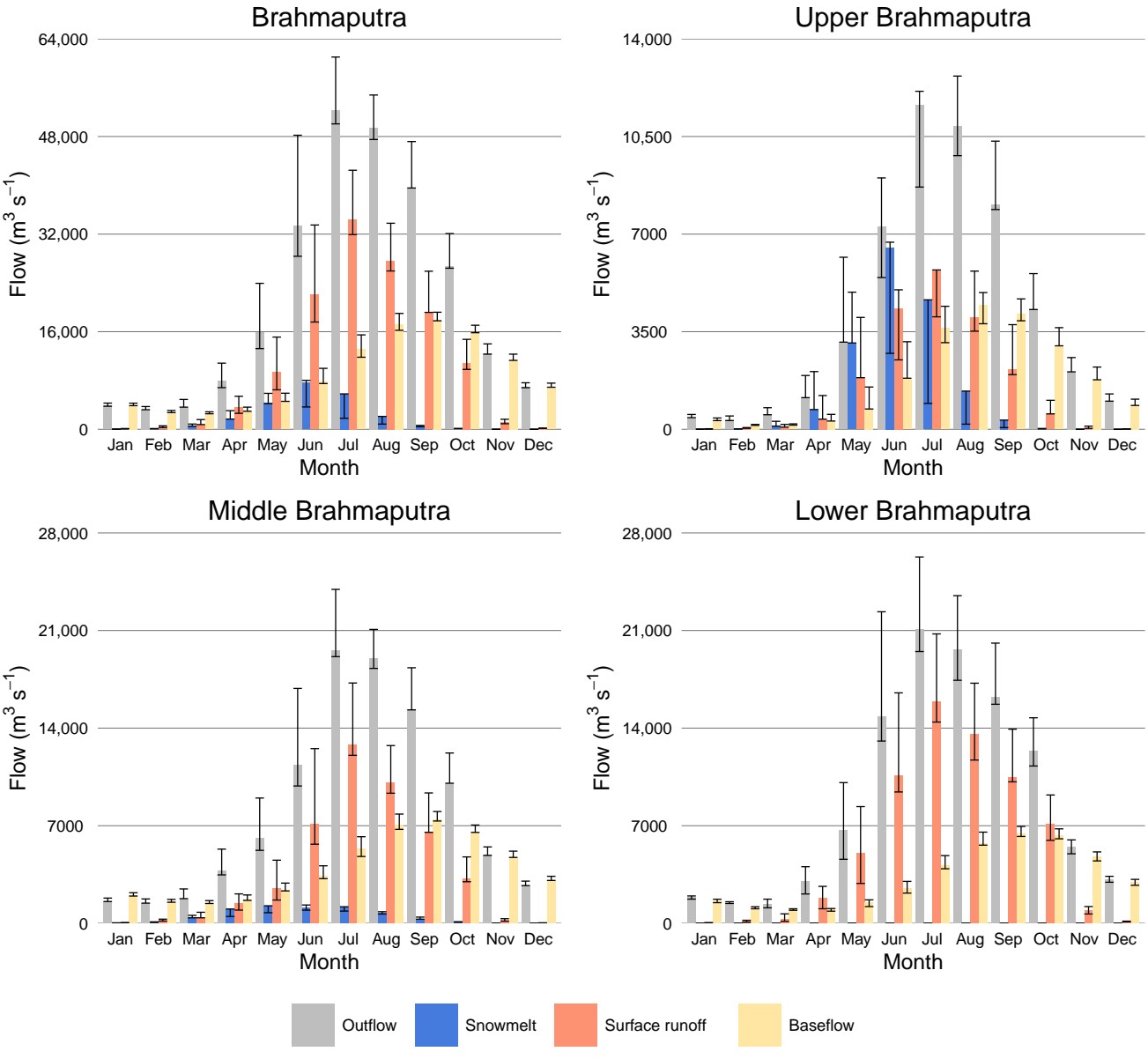

**Figure 12.** Monthly average contribution of three key runoff components (bars) to outflow along with snow and glacier-melt in the whole Brahmaputra basin and three regions; error bars show the range of values (maximum and minimum) projected under the four climate scenarios.

The snowmelt contribution to total runoff was estimated at 1253 $m^3s^{-1}$, or 6% of the total annual outflow of the Brahmaputra observed at Bahadurabad. The actual amount of snowmelt is much higher (1911 $m^3s^{-1}$), but approximately one-third is lost in the processes of refreezing of snow, evaporation, and percolation to deep aquifers and does not contribute to streamflow either as surface runoff or baseflow. The seasonal contribution of snowmelt to the total flow of the whole Brahmaputra also varies with 2.1% in winter and dry season flow, 11% in pre-monsoon, 6.8% in monsoon, and 0.4% in post-monsoon (Table 7). Note that there is a slight difference between AWY in the basin (22,829 $m^3s^{-1}$) and total outflow at Bahadurabad (22,162 $m^3s^{-1}$) due to loss from various processes during flow routing from upstream to downstream.

The snowmelt contribution to annual flow in the upper Brahmaputra was 21%, the highest proportion in the three regions of the river basin (Table 6). Considering the present study's snowmelt estimation only, 21% snowmelt contribution is consistent with the value of 25% for snow and glacier melt together reported by Immerzeel et al. [62] and Lutz and Immerzeel [51] in the same region of the river basin. The snowmelt contributed 0.5%, to the total outflow from the upper Brahmaputra in the winter and dry season, 31% pre-monsoon, and 24% in the monsoon (Table 7, Figure 11). On the other hand, the snowmelt contributes only 5% to annual flow in the middle Brahmaputra (4.8% in the winter and dry, 13.2% pre-monsoon, and 4.3% in the monsoon) and nothing in the lower Brahmaputra (Tables 6 and 7; Figure 12). Among the middle Brahmaputra watersheds, the highest contribution of snowmelt was in the Dibang, Lohit, and upper Subansiri watersheds, with a very small contribution in the Kameng, Manas, Sunkosh, and Teesta watersheds, and no contribution from snowmelt in the remainder (Table 6). The percentage contribution of flow components in the different seasons was determined relative to seasonal water yield (Figure 11), but the percentage values don't indicate the actual volume involved. For example, even though the relative contribution of snowmelt during the monsoon is lower than in the pre-monsoon, the total snowmelt amount in the monsoon is higher (Table 7). The seasonal analysis showed that the primary snow melting season in the Brahmaputra basin is pre-monsoon (April–May) and monsoon (June–September), this is consistent with the findings of Panday et al. [12], who identified May–October as the main snowmelt season in the Hindu-Kush Himalayan river basins.

*4.2. Impact of Climate Change*

The projected annual average climate change impact in 2050 in the Brahmaputra River Basin was calculated using two CC projections (RCP 4.5 and RCP 8.5) with four climatic conditions for each (total eight scenarios). The results are shown in Figure 13. The impact on the hydrology showed similar trends under all scenarios but with differing magnitudes. The temperature increase in most parts of the basin will have a direct impact on evapotranspiration. Annual average actual evapotranspiration (AET) is projected to go up by 8% on average under both scenarios (5–9% under RCP 4.5 and 6–11% under RCP 8.5). The increase in AET will, in turn, have a negative impact on soil moisture content. SWAT model projected a reduction in basin-wide average soil moisture by 4% (3–5%) under RCP 4.5 and 5% (2–8%) under RCP 8.5, with a greater decrease in the upper Brahmaputra (8–9%) than in the middle Brahmaputra (2%) or lower Brahmaputra (1%).

The annual average basin-wide precipitation is projected to increase by 8% on average (range 4–9%) under RCP 4.5 and 7% on average (1–18%) under RCP 8.5 (Figure 13). The increase in precipitation will lead to a rise in water yield. The annual water yield (AWY) is projected to increase by 8% (5–11% under RCP 4.5 and 0–21% under RCP 8.5) with the maximum average change in the pre-monsoon season (16% under RCP 4.5 and 29% under RCP 8.5), followed by the dry season (8% under RCP 4.5 and 7% under RCP 8.5), the monsoon season (8% under RCP 4.5 and 6% under RCP 8.5), and the post-monsoon (7% under RCP 4.5 and 4% under RCP 8.5) (Figures 14 and 15; the ranges projected for the different scenarios under each projection are given in the table).

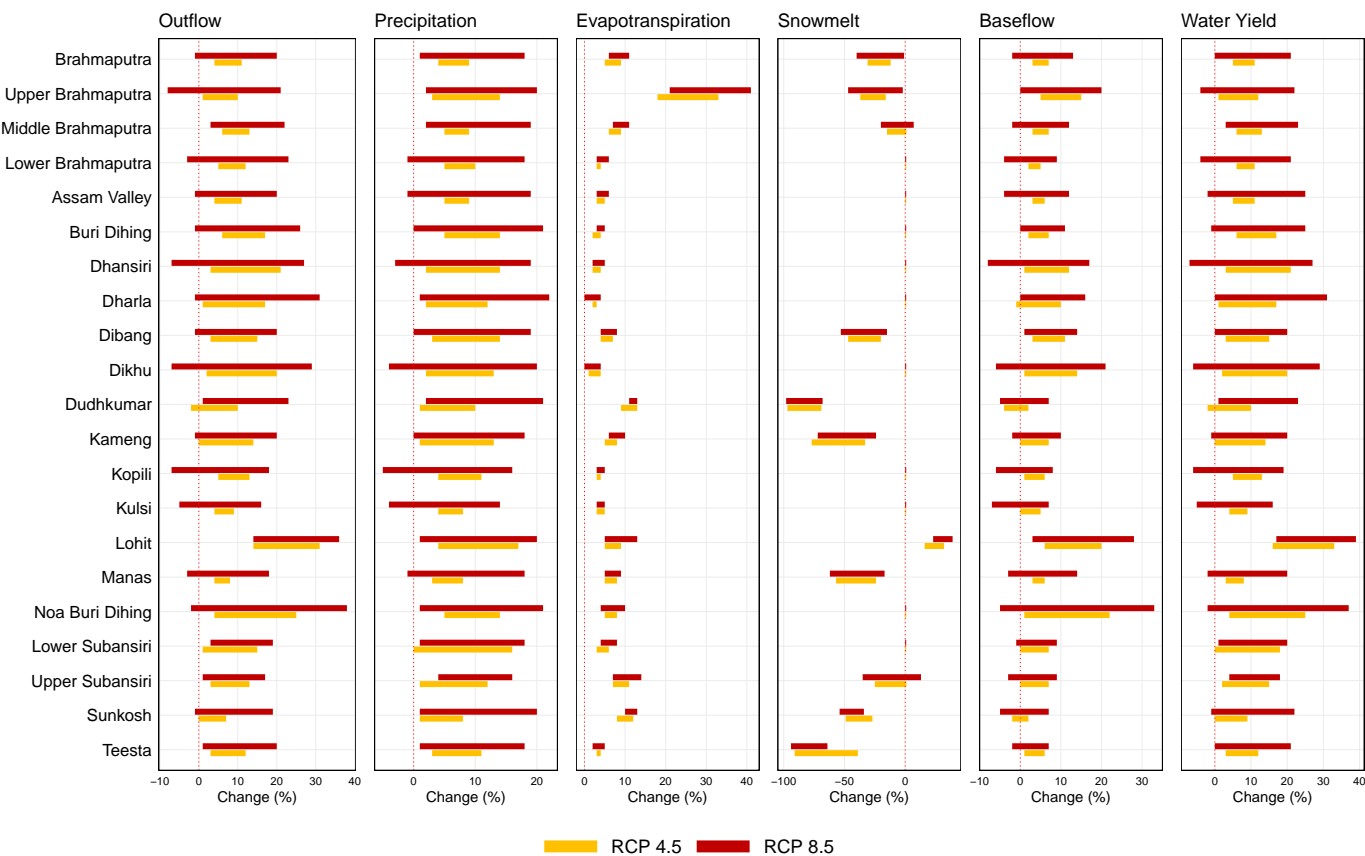

**Figure 13.** Annual maximum and minimum percentage changes in key water balance components for the entire Brahmaputra basin and its large subbasins under RCP4.5 and RCP8.5 scenarios. For each RCP, maximum and minimum ranges are based on four climatic conditions.

The AWY is projected to increase by 9% (1–12%) in the upper Brahmaputra under RCP 4.5 and by 8% (−4–22%) under RCP 8.5, with the maximum average change in the pre-monsoon season (46% under RCP 4.5 and 64% under RCP 8.5), followed by the winter and dry (18% under RCP 4.5 and 22% under RCP 8.5), post-monsoon (17% under RCP 4.5 and 18% under RCP 8.5), and monsoon season (−2% under RCP 4.5 and 0% under RCP 8.5) (Figures 14 and 15). In the middle Brahmaputra, the AWY is projected to increase by 9% under both RCP 4.5 (6–13%) and RCP 8.5 (3–23%), with the maximum average change in the pre-monsoon season (12% under RCP 4.5 and 24% under RCP 8.5), followed by the winter and dry (9% under both RCP 4.5 and RCP 8.5), monsoon (10% under RCP 4.5 and 8% under RCP 8.5), and post-monsoon season (8% under RCP 4.5 and 6% under RCP 8.5).

Among the different hydrological mechanisms, snowmelt is the process projected to be most affected by CC in the Brahmaputra basin. Seven out of eight selected climate conditions in this study are projected to be warmer and wetter than the base period except for the 'warm and dry' condition (Table 3) which could lead to an increase in snowmelt processes. The results of the seasonal analysis of this study support this idea. Basin-wide snowmelt was projected to increase by 44% on average under RCP 4.5 (26–62%) and 38% under RCP 8.5 (30–46%) during the winter and dry season, 18% (6–40%) under RCP 4.5 and 22% (1–48%) under RCP 8.5 during pre-monsoon, and 6% (−6–13%) under RCP 4.5 and 11% (1–24%) under RCP 8.5 during the post-monsoon season (Figures 14 and 15).

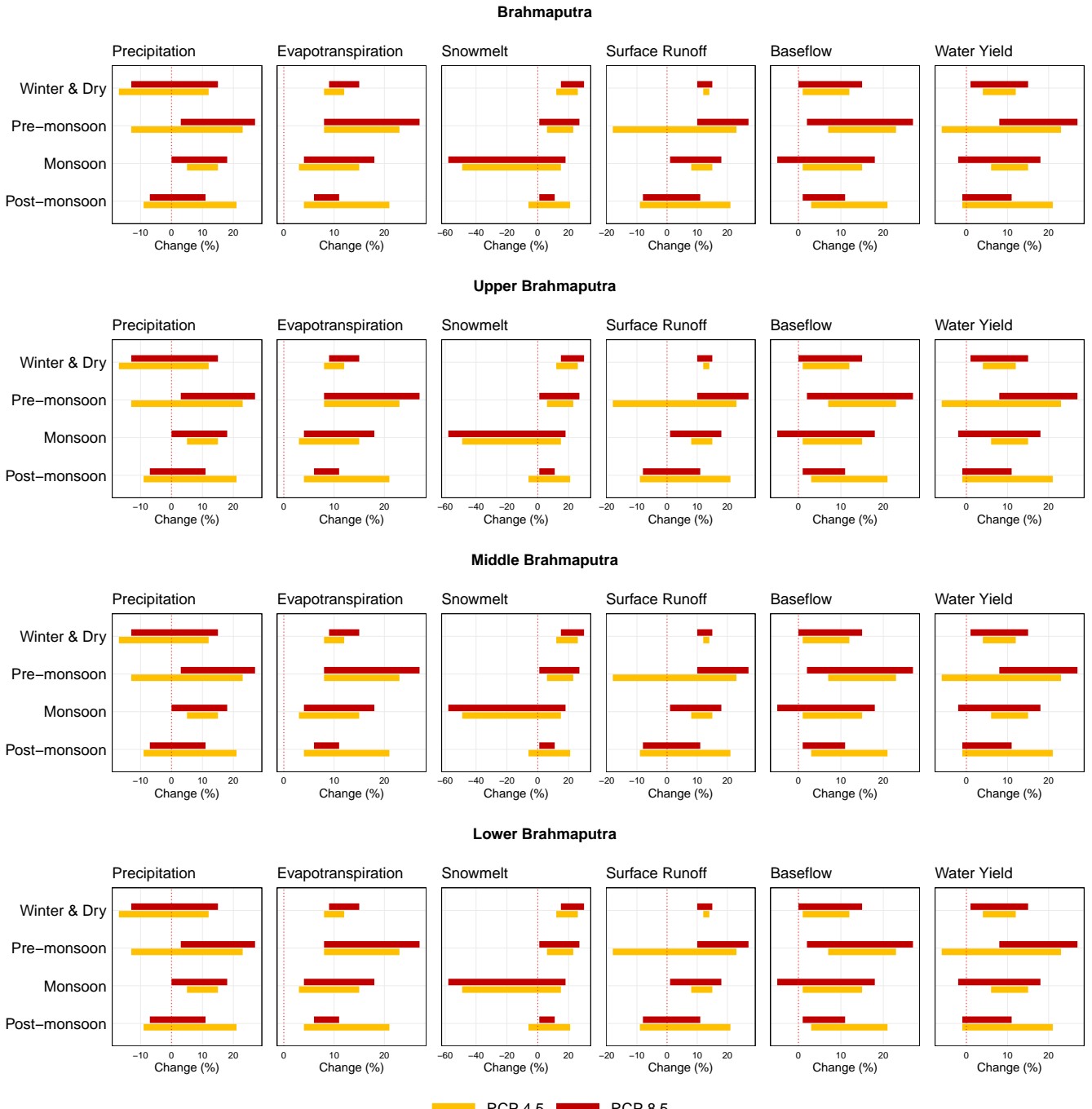

**Figure 14.** Seasonal maximum and minimum percentage changes in key water balance components for the entire Brahmaputra basin and its three major regions (Upper, Middle and Lower) under RCP4.5 and RCP8.5 scenarios. For each RCP, maximum and minimum ranges are based on four climatic conditions.

However, despite the increase in snowmelt in all the non-monsoon periods, the total average annual melt is expected to decrease by 17% under both RCP 4.5 (12–31%) and 8.5 scenarios (1–40%) (Figure 13) which is attributed to a substantial decline during the monsoon season. The monsoon snowmelt is projected to decrease by 33% (23–49%) under RCP 4.5 and 34% (7–58%) under RCP 8.5 (Figure 14). As the monsoon is the main snowmelt season, with about 71% of the total annual snowmelt (Table 7), the reduction in monsoon snowmelt leads to a net reduction in total annual snowmelt over the Brahmaputra basin. The warmer and wetter climate could also lead to a decline in the amount of snowfall,

creating conditions under which there is less snow cover in the high-altitude areas at the start of the monsoon, and resulting in less snowmelt in the monsoon. The decrease in monsoon melt projected by our model supports this hypothesis. The decrease in monsoon snowmelt not only balances out the snowmelt increase in the non-monsoon periods but reduces the annual snowmelt by a large amount under both CC scenarios.

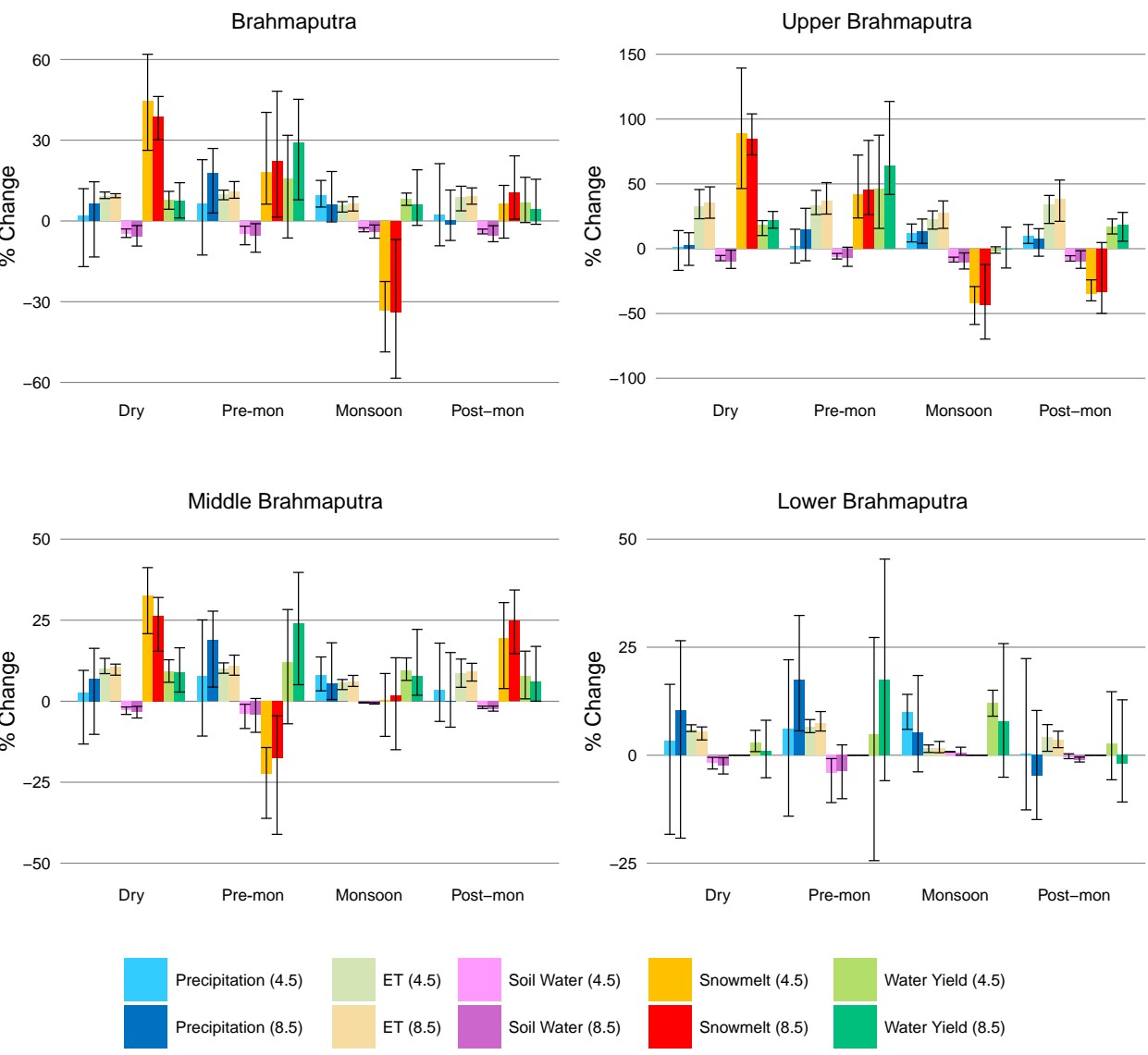

**Figure 15.** Season-wise average impact of climate change on five key water balance components (bars) in the whole Brahmaputra basin and three regions; error bars show the range of values (maximum and minimum) projected under the four climate scenarios.

Within the basin, the upper Brahmaputra is likely to be most affected by CC. Despite the projected increase in overall water yield in all seasons, the annual snowmelt is expected to decrease by an average value of 22% under both scenarios (18–33% under RCP 4.5 and 21–41% under RCP 8.5) (Figure 13). Immerzeel et al. [62] and Lutz and Immerzeel [51] projected a 19.6% decrease in annual snow and glacier melt for the upper Brahmaputra, which is within the range of our findings. Their results also showed that while the share of meltwater is projected to drop, the rain runoff contribution to the total runoff and flow will most likely increase in this part of the Brahmaputra basin. Our present study also supports this. Despite the net reduction in snowmelt over the year, snowmelt was projected to increase by 89% and 85% during the winter and dry season, and 42% and

45% in the pre-monsoon respectively under RCP 4.5 and RCP 8.5 (Figure 14 with ranges; Figure 15). However, the projected reduction in monsoon snowmelt by 42% and 43% and post-monsoon snowmelt by 35% and 33% respectively under RCP 4.5 and RCP 8.5 outweighs the increase in the other seasons.

In the middle Brahmaputra, which includes the snow and glacier-fed watersheds in the southern Himalayas, the average annual snowmelt is expected to decrease by 5% and 3% respectively under RCP 4.5 and RCP 8.5 (Figure 13). The snowmelt is also expected to decrease in the pre-monsoon season by 22% and 18%, but to increase by 33% and 26% in the winter and dry season, and 19% and 25% in post-monsoon respectively under RCP 4.5 and RCP 8.5 (Figures 14 and 15). However, changes calculated under the eight different scenarios for monsoon melt vary widely from a decrease of 11% to an increase of 9% under RCP 4.5 and a decrease of 15% to an increase of 13% under RCP 8.5, averaging out at 0 and 2%, respectively. These changes reflect the combined effect of the projected temperature rise, reduced snowfall, and increased snowmelt. The change in the middle Brahmaputra snowmelt in the monsoon is probably not significant and does not affect the projected water yield substantially. Overall, the increase in post-monsoon and winter and dry season water yield in the middle Brahmaputra is probably due to an increase in both precipitation and snowmelt, whereas the increase in water yield in the pre-monsoon and monsoon seasons is due to the increase in rainfall only.

The part of the Brahmaputra basin projected to be least affected by CC is the lower Brahmaputra, which includes watersheds such as the Assam Valley, Dharla, Dikhu, Kopili, Kulsi, Noa Buri Dihing, and lower Subansiri. The AWY is projected to increase by 9% under RCP 4.5 (6–11%) and 6% under RCP 8.5 (−4–21%), with the maximum average change in the monsoon season (10% under RCP 4.5 and 8% under RCP 8.5) and pre-monsoon season (5% under RCP 4.5 and 18% under RCP 8.5), and the least in the winter and dry (3% under RCP 4.5 and 1% under RCP 8.5) and post-monsoon season (3% under RCP 4.5 and −2% under RCP 8.5) (Figures 14 and 15; see table for the range of values projected). Figure 16 shows the spatial distribution of key components of the Brahmaputra hydrology under base conditions and the average percentage change under RCP 4.5 and 8.5 scenarios.

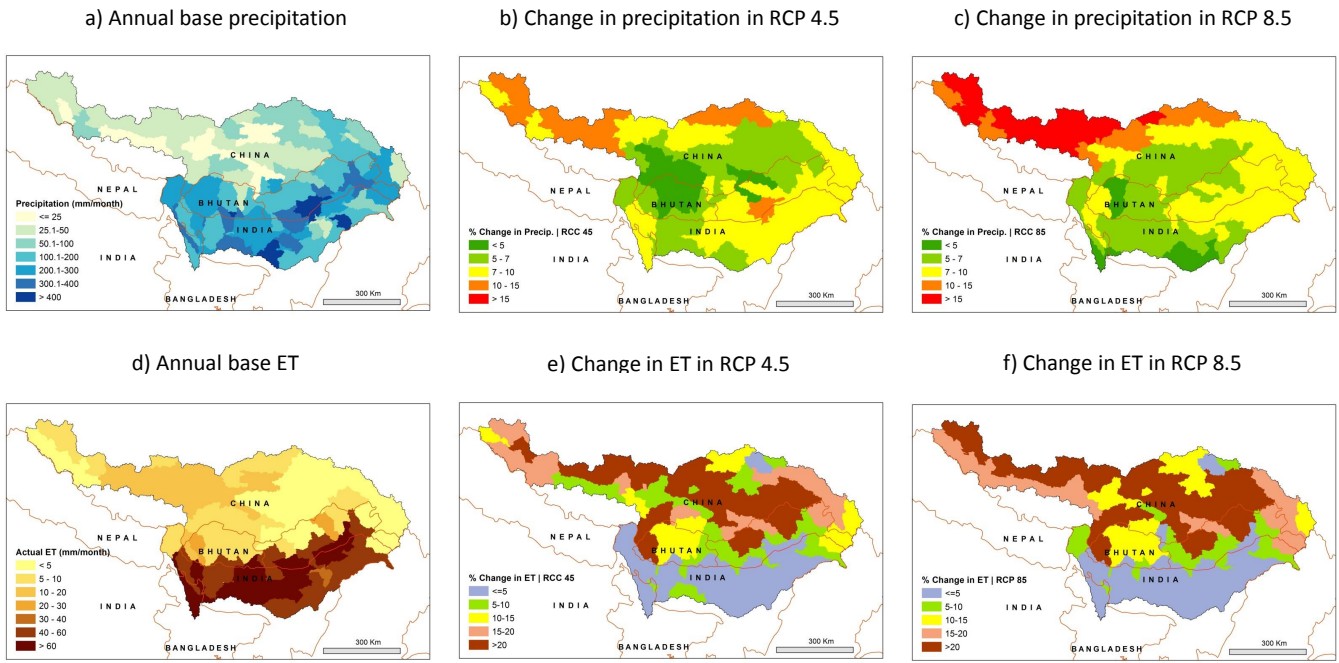

**Figure 16.** *Cont*.

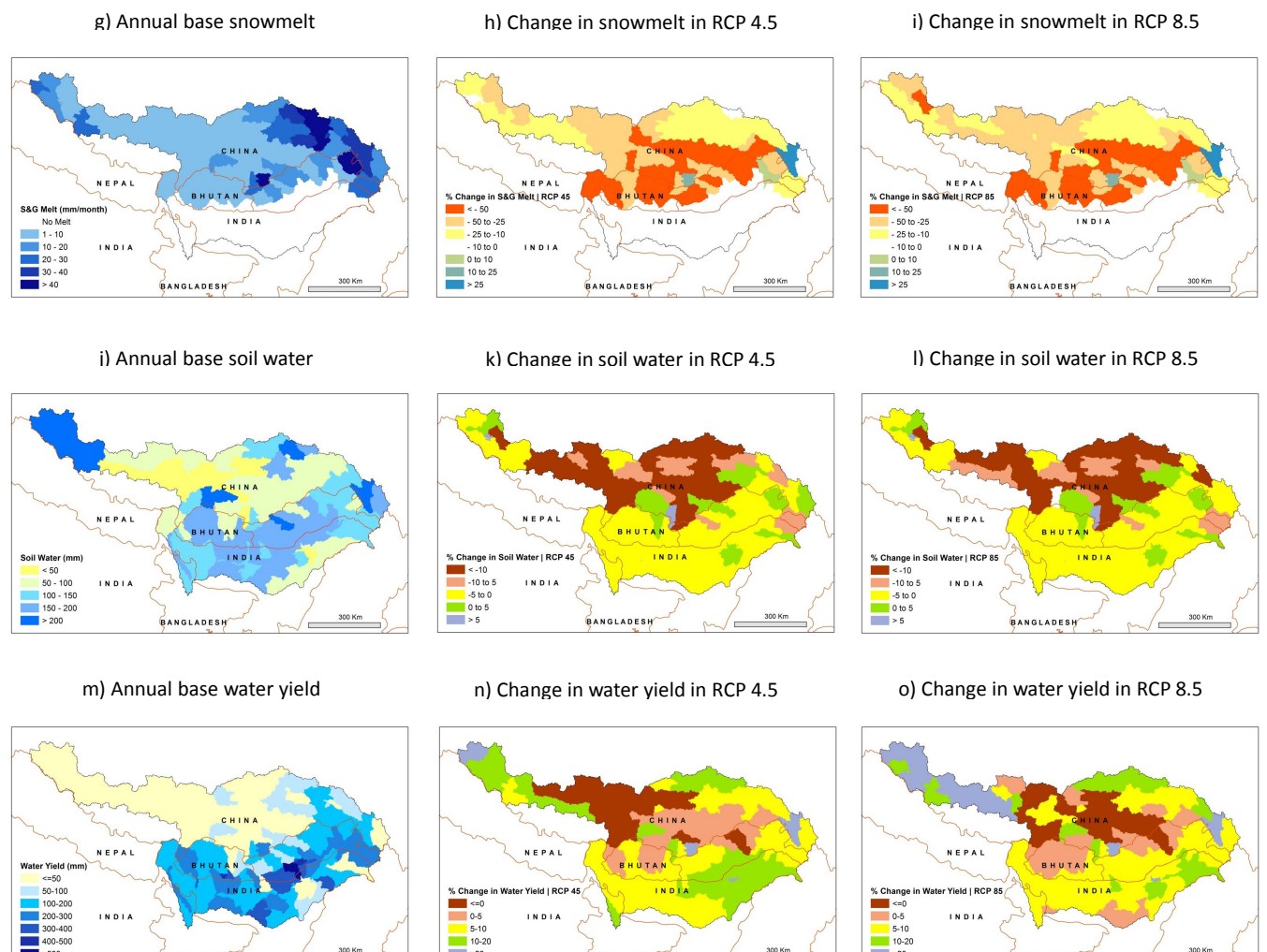

**Figure 16.** Spatial distribution of water balance components under the base condition and impact of climate change under the RCP 4.5 and 8.5 scenarios.

## 5. Conclusions

### 5.1. Summary of Findings

This paper presents the annual and seasonal water balance for the Brahmaputra basin as a whole and its three major regions (i.e., upper, middle, and lower) calculated using simulations of a SWAT model developed for the basin. The paper also presents the annual water balance for 16 individual watersheds in the middle and lower Brahmaputra and Assam floodplains as demarcated in Figure 2. The model was also used to project the impact in the mid-twenty-first century of two climate change (CC) projection scenarios (RCP 4.5 and 8.5) using four climatic conditions. Considering the limited availability of observed flow data for the basin, the calibration and validation results for the model and its overall accuracy in simulating monthly, seasonal, and annual basin hydrology, and snowmelt processes were reasonable and sufficient for the model to be used in a water balance study of the basin. The CC change scenarios project an increase in temperature, which is expected to have an impact on seasonal snowfall and snowmelt processes, as well as an increase in precipitation.

The CC impact assessment suggests that the upper Brahmaputra, in the Tibetan Plateau, will be the part of the river basin most affected by CC. The upper Brahmaputra contributes about 20% of the average annual water yield (AWY) in the basin, which is expected to increase by 5% and 9% on average across the different climatic conditions under RCP 4.5 and 8.5, respectively. At present, snowmelt contributes about 21% of the region's

AWY, the proportional contribution is projected to drop by 22% under both scenarios, but the overall AWY is projected to increase as a result of the increase in precipitation. The contribution of snowmelt in the non-monsoon seasons (winter and dry, pre-monsoon, and post-monsoon) is projected to increase; therefore, the projected net reduction in annual snowmelt is coming from a significant reduction during the monsoon season.

The middle Brahmaputra includes all the snow and glacier watersheds on the southern Himalayan slope and is expected to be the second most affected region. This region contributes about 42% of Brahmaputra AWY, which is expected to increase by 9% on average across the different climatic conditions under both RCP 4.5 and RCP 8.5. At present, snowmelt contributes about 5% of the region's AWY, which is projected to drop by 5% and 3% on average under the RCP 4.5 and 8.5 scenarios, respectively. In contrast to the upper basin, the projected decline in annual snowmelt contribution is the result of a considerable reduction in the pre-monsoon season.

The large decline in monsoon snowmelt in the upper Brahmaputra and pre-monsoon snowmelt in the middle Brahmaputra can be linked to the projected temperature rise in these two regions. For example, the temperature rises during the winter and dry and pre-monsoon seasons in the upper Brahmaputra will lead both to an increase in the snowmelt rate and a reduction in snowfall, leaving less snow available to melt on the mountain tops during the monsoon. The reduced snow cover in the monsoon may also induce more glacier melt in the region during the monsoon season. However, the present study was unable to access that because of our model's inability to simulate glacier melt. As mentioned earlier, SWAT in its default settings can simulate snowmelt only. Nevertheless, the decline in monsoon snowmelt is projected to lead to an overall decline in annual snowmelt in the upper Brahmaputra. The projected increase in temperature in the winter and dry seasons could create the same conditions in the middle Brahmaputra in the pre-monsoon season with both a high snowmelt rate and less snowfall during the winter and dry season leaving less snow available to melt during the pre-monsoon season, leading to a decline in both pre-monsoon and monsoon meltwater. Overall, the share of meltwater is expected to decrease, and the share of rain runoff to increase and become more dominant in the hydrology of both the upper and middle Brahmaputra due to CC.

The lower Brahmaputra is expected to be the part of the basin least affected by CC. The lower Brahmaputra contributes about 38% of the Brahmaputra AWY, and this contribution is expected to increase by 9% under RCP 4.5 or 6% under RCP 8.5. The AWY of the whole Brahmaputra basin is projected to increase by 8% under both scenarios. Despite an increase in snowmelt in the non-monsoon seasons in the upper Brahmaputra and in the winter and dry and post-monsoon seasons in the middle Brahmaputra, the net AWY of the whole Brahmaputra is expected to drop by 17% on average across the different climatic conditions under both scenarios.

## 5.2. Contribution to Basin Water Management

This paper provides a detailed hydrological assessment of the high-altitude snow and glacier watersheds and low-altitude rainfed watersheds of the Brahmaputra River Basin under current hydro-climatology and future climate change conditions. It is imperative to carry out monthly and seasonal analyses for a river basin such as the Brahmaputra with extreme hydro-climatological seasonality in order to calculate the actual water account of the basin. According to our best knowledge, this paper is the first hydrological study to provide a detailed account of the Brahmaputra River Basin with three major regional separations and 16 major watersheds on the southern Himalayan slopes. We believe this study will be valuable not for only those in government institutions but also other organizations and stakeholders within the river basin, such as community and indigenous rights groups, conservationists, non-government agencies, fishers, farmers, and others.

In recent times, several initiatives have been taken in the basin to exploit water availability, particularly for hydroelectricity production. There are 12 hydro dams in operation in the upper Brahmaputra in China, eight in the middle Brahmaputra in Bhutan, and 19 in the middle

and lower Brahmaputra in India [63]. The exact numbers of large dams planned and under construction are difficult to find, but a map by Alley et al. [64] indicates that the number could be at least 30. When small dams are included, the total number is likely to be many more. These hydro dams are usually runoff dams, which are often considered to be less harmful to the downstream region in terms of flow availability. However, there is growing evidence to suggest that both small and large dams have detrimental effects on river flow and ecology, the environment, community livelihoods, food security, and health [64,65]. The large numbers of current and proposed dams in the basin are also likely to create challenges related to water availability in the lower riparian countries [66]. Our study can help support project planning and design of these dams, taking into account the uncertainty in the hydrology and potential future changes, as indicated by the results of the model simulations. The analysis can also help basin-wide water management with the participation of all stakeholders to ensure equity and sustainability [67] and can help non-government stakeholders to understand the basin's water availability and to anticipate challenges and opportunities in the coming years so that they are well informed and better prepared for water management negotiations and can suggest alternative options.

*5.3. Limitations*

The major limitations of the present study lie with the availability of observed streamflow data and the SWAT model's default capability. We faced difficulty in collecting observed streamflow data and the choice of data to be utilized for model calibration and validation. It is well known that the more observed data is available, the chance of developing a more accurate model is increased, and we were lacking behind on this ground. In addition, the default SWAT setup cannot simulate snow and glacier melt both, rather it simulates snowmelt only. As a result, we were unable to report the glacier melt contribution to the basin's base water yield or to assess the CC impact on it. This is indeed a major limitation for a study aiming to understand the snow and glacier hydrology of a river basin such as the present one. Moreover, we ignored the future possible change in land use and land cover including the snow cover retreat in high-altitude mountain areas of the basin in our CC SWAT simulation. Applying continuous change in basin physiographic conditions in the SWAT model is beyond its current ability. We did not also include currently operating hydro dams in the basin or groundwater irrigation practices in the Assam Valley. Hydro dams alter the natural variability of the streamflow while groundwater irrigation can affect baseflow in the non-monsoon months. The overall effects of these water projects on streamflow are understood to be not very significant at present, hence we opted for not including those dam and irrigation practices in the present SWAT model.

**Author Contributions:** W.P., S.R.B., A.B.S. and S.W. conceptualized the research study, designed the method, and wrote the original draft, as well as prepared, reviewed, and edited the manuscript. The data preparation and analysis were done by W.P., S.R.B., M.S.H., T.K.M. and L.C.M., as well as the model calibration and validation, and they also contributed to the writing for the evaluation of the SWAT model. All authors have read and agreed to the published version of the manuscript.

**Funding:** This study was funded by the HICAP project. It was also partially supported by core funds of the International Centre for Integrated Mountain Development (ICIMOD).

**Data Availability Statement:** The data presented in this study will be made available by the authors upon request.

**Acknowledgments:** This study was undertaken under the Himalayan CC Adaptation Programme (HICAP), which is implemented jointly by ICIMOD, CICERO, and GRID-Arendal in collaboration with local partners and is supported by the Governments of Norway and Sweden. The views and interpretations in this publication are those of the authors. They are not necessarily attributable to ICIMOD and do not imply the expression of any opinion by ICIMOD concerning the delimitation of frontiers or boundaries. The authors would like to thank Ms Yedda Wang, for her encouragement and support to submit the manuscript.

**Conflicts of Interest:** The authors declare no conflict of interest.

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
