# Peer review of "Climate Change Impacts on the Hydrology of the Brahmaputra River Basin"

_climate, doi:10.3390/cli11010018_

Round 1

Reviewer 1 Report

The authors present a study detailing the impacts of climate change on the hydrology of the Brahmaputra River Basin in the Himalayas. The SWAT model was used to detail the water balance of this river basin under several climate change scenarios. The article is well written, figures are well put together, and appropriate techniques were applied for model validation. This paper is novel in that the hydrology of the region has not been evaluated previously for climate change impacts in this manner. I feel that the article will be acceptable for publication, once some minor issues are addressed. I feel that text of the manuscript does not need to be revised other than a final proof reading upon resubmission to correct minor typos. However, Tables 8 and 9 are dense and difficult to interpret, Table 9 specifically. These tables need to be improved for legibility. Once these issues are addressed, no other changes are requested by me. 

Reviewer 2 Report

The manuscript titled “Climate change impacts on the hydrology of the Brahmaputra River Basin” present annual and seasonal water balance for such a basin and some its sub-basin, under selected scenarios and by using the Soil and Water Assessment Tool (SWAT). It is well written and structured and does not require major edits.

 However it is too long in some parts, starting with the Introduction. Some of them could be reduced or moved to other sections. For example the second part of the last paragraph of page 2, from “ The ArcSWAT 2012 version …” to the end, may be modified and placed at the beginning of the Section 2.

 The citation style is not that of MDPI. “References must be numbered in order of appearance in the text (including table captions and figure legends) and listed individually at the end of the manuscript” as the Instruction for Authors requires. References must be described as the instruction shows and depending on the type of work.

 The finding of the manuscript depend on the correctness of the prediction of the temperature increase. If the projected expectation were wrong, the study would have been useless.

 On the other hand, the impacts should be of different degree in the three regional separations and major watersheds. I suggest adding a final summary table illustrating these impacts and their consequences. However, the authors state that their study “study can help support project planning and design … water management negotiations and can suggest alternative options” (page 28). Could the authors be more specific on this point?
